# In situ mechanical reinforcement of polymer hydrogels via metal-coordinated crosslink mineralization

Sungjin Kim [1,4], Abigail U. Regitsky[1,4], Jake Song [1], Jan Ilavsky[2], Gareth H. McKinley [3] & Niels Holten-Andersen [1✉]

Biological organic-inorganic materials remain a popular source of inspiration for bioinspired materials design and engineering. Inspired by the self-assembling metal-reinforced mussel holdfast threads, we tested if metal-coordinate polymer networks can be utilized as simple composite scaffolds for direct in situ crosslink mineralization. Starting with aqueous solutions of polymers end-functionalized with metal-coordinating ligands of catechol or histidine, here we show that inter-molecular metal-ion coordination complexes can serve as mineral nucleation sites, whereby significant mechanical reinforcement is achieved upon nanoscale particle growth directly at the metal-coordinate network crosslink sites.

[1] Department of Materials Science and Engineering, Massachusetts Institute of Technology, Cambridge, MA, USA. [2] X-ray Science Division, Advanced Photon Source, Argonne National Laboratory, Lemont, IL, USA. [3] Department of Mechanical Engineering, Massachusetts Institute of Technology, Cambridge, MA, USA. [4] These authors contributed equally: Sungjin Kim, Abigail U. Regitsky. ✉email: holten@mit.edu

From abalone shells to tendons, nature displays a wide array of organic–inorganic composites with a broad spectrum of mechanical properties spanning from hard and fracture resistant to soft and tough[1]. This remarkable variety is in large part thanks to the evolution of cell-orchestrated material assembly processes, in which cascades of catalytic proteins and ion transport mechanisms coordinate the templated mineralization of macromolecular scaffolds[2–5]. In the absence of successful mimicry of such active cellular material assembly, various biomacromolecular hydrogel networks have been employed in attempts to gain some control over material mineralization processes, and different biomolecular functionalities have been utilized to bind metal ions, upon which metal oxide minerals can nucleate and grow[6–9]. In an attempt to gain deeper insights on the coupling between early stage mineral growth dynamics and polymer network mechanics, we sought to nucleate metal oxide particle growth directly at chain-end inter-molecular crosslinking sites in a model hydrogel scaffold. To assemble such a system, we took inspiration from the incorporation of Fe-catechol coordinate crosslinks in mussel holdfast threads, a material design principle, which have been widely utilized in the integration of tunable stimuli-responsive motifs in advanced hydrogel engineering[10–12]. Since catechol ligands have also been shown to bind strongly to iron oxide nanoparticle surfaces[13–18], we hypothesized that a catechol-polymer network could be utilized as a simple mineralization scaffold, wherein in situ iron oxide mineral nucleation is targeted directly to the network crosslink sites by strong coordination bonding, thereby avoiding the compromise in network elasticity normally observed upon particle incorporation[15,19–22]. While the Fe-catechol system serves as an ideal platform to test our proof of concept of metal-coordinate crosslink mineralization[15,23–25], we further examined if this approach could be extended to different metal-ligand coordination systems using histidine-modified polymers, which bind ions and minerals of nickel or copper.

Here, we introduce our findings supporting that metal-ion coordination complexes can indeed serve as direct mineral nucleation sites, whereby significant mechanical reinforcement is achieved upon nanoscale particle growth directly at the metal-coordinate network crosslink sites.

## Results

**In situ metal-coordinate crosslink mineralization approach.** As a scaffold for the in situ mineralization of iron oxide nanoparticles, we employed a hydrogel composed of catechol-functionalized 4-arm polyethylene glycol (4cPEG) polymers crosslinked by permanent catechol-catechol covalent bonds and transient $Fe^{3+}$-catechol metal-coordinate complexes (labeled in Fig. 1a as Mineral-Free). This initial Mineral-Free polymer hydrogel scaffold was established according to a mussel-inspired gel assembly method[11], which introduces both covalent and coordination network crosslinks by mixing 4cPEG and $Fe^{3+}$ at a 1:1 molar ratio of metal-ion to catechol ligand, followed by raising the pH to 12 with $OH^-$. To subsequently initiate in situ mineralization of $Fe_3O_4$ within this polymer scaffold (labeled as In Situ in Fig. 1b), we added just enough $Fe^{2+}$ to the gel to hypothetically form a single formula unit of $Fe_3O_4$ at each possible *tris*-catechol-coordination crosslink site (please see "Methods" section for details). To test if this targeted crosslink mineralization results in enhanced network reinforcement as predicted, we prepared two control polymer networks; in Ex Situ networks (Fig. 1c) we initiated nucleation and growth of $Fe_3O_4$ particles outside of the metal-coordinate polymer network before mixing with the 4cPEG polymer. In Ligand-Free samples, while following the same sample protocol as In Situ gels, we used a polymer with no metal-

coordinating ligand, whereby any crosslink mineralization should be lost (labeled as Ligand-Free in Fig. 1d).

We used small amplitude oscillatory shear (SAOS) rheology to measure the storage moduli ($G'$) and loss moduli ($G''$) of the Mineral-Free and mineralized networks. As demonstrated in past work, SAOS is a useful tool to study both in situ mineralization in hydrogels[26] and the mechanical differences between 4cPEG gels crosslinked via $Fe^{3+}$ ions or $Fe_3O_4$ nanoparticles[15]. As shown in Fig. 1e–f, we observed at least a threefold increase in plateau modulus ($G_p$) of the In Situ mineralized gels compared to all other networks (see Supplementary Fig. 1 for multiple datasets). To test if mineralization directly targeted at the metal-coordinate crosslink sites results in enhanced mechanical coupling of $Fe_3O_4$ particles within the polymer network, we can compare our data with predictions from the Guth-Gold equation for non-interacting fillers in rubbers[27] and gels[28]. With the assumption of complete mineralization of all reactants, and thereby a maximum mineral volume fraction ($\Phi_{mineral}$) of 0.074 vol%, Guth-Gold estimation predicts that the increase in stiffness (as captured by the plateau modulus-$G_p$) of the Mineral-Free gel network expected upon incorporating such small amount of non-interacting particles is only around 10 Pa (Supplementary Fig. 2). Instead, the stiffness increase observed upon the in situ gel mineralization is over three orders of magnitude higher (around 30,000 Pa), which supports that the minerals formed are strongly interacting with the network, in agreement with metal-coordinate crosslink mineralization. In contrast, with uncontrolled mineral nucleation and growth initially outside the metal-coordinating network, Ex Situ gels displayed comparatively poor mechanical properties, likely due to limited network incorporation of the pre-formed large and heterogeneous particles visibly precipitating in the sample as shown in Fig. 1c. Note that the Ex Situ gel is formed by mixing a solution of pre-formed mineral precipitates (as in Mineral-Only, further characterized in Supplementary Fig. 3) with the polymer solution to induce gelation. Finally, within solutions of polymers without a metal-coordinating ligand on the backbone (Ligand-Free), as shown in Fig. 1d we observed even more pronounced precipitation compared to Ex Situ, in addition to no gel assembly upon mineralization. These observations combined support a direct role of metal-coordination in controlling crosslink mineralization as well as negligible contributions from mineral-polymer interactions outside the network crosslink sites to both network formation and elasticity.

To further explore the role of metal-coordination in targeted crosslink mineralization, we obtained Raman spectra of the In Situ, Ex Situ, and Mineral-Free networks, shown in Fig. 1g alongside reference spectra of pure 4cPEG (Polymer-Only) and polymer-free iron oxide minerals (Mineral-Only). The resonance peaks observed between ~530 to ~650 cm$^{-1}$ in the In Situ, Ex Situ, and Mineral-Free gels confirm the presence of Fe-catechol coordinate interactions. Specifically, the three distinct peaks at 531, 593, and 643 cm$^{-1}$ observed in the spectrum of the Mineral-Free gel have been previously assigned to the coordination bonding interactions between $Fe^{3+}$ and the oxygens of the catechol ligand in a high pH environment[10]. The peak broadening observed in the In Situ gel spectrum is reminiscent of previous results from composite $Fe_3O_4$ nanoparticle coordinate crosslinked 4cPEG gels reported by Li et al.[15]. These observations support the hypothesis that in situ gel mineralization can indeed be targeted at the gel network crosslinking sites through a transformation of initial $Fe^{3+}$-catechol coordinate complexes into catechol-bound $Fe_3O_4$ mineral particle crosslink structures. In addition, the spectrum from the Mineral-Only sample displays a peak centered around 680 cm$^{-1}$ corresponding to $Fe_3O_4$ (magnetite) minerals[24,25], a peak which also appears in the spectrum of In Situ samples alongside a smaller peak around

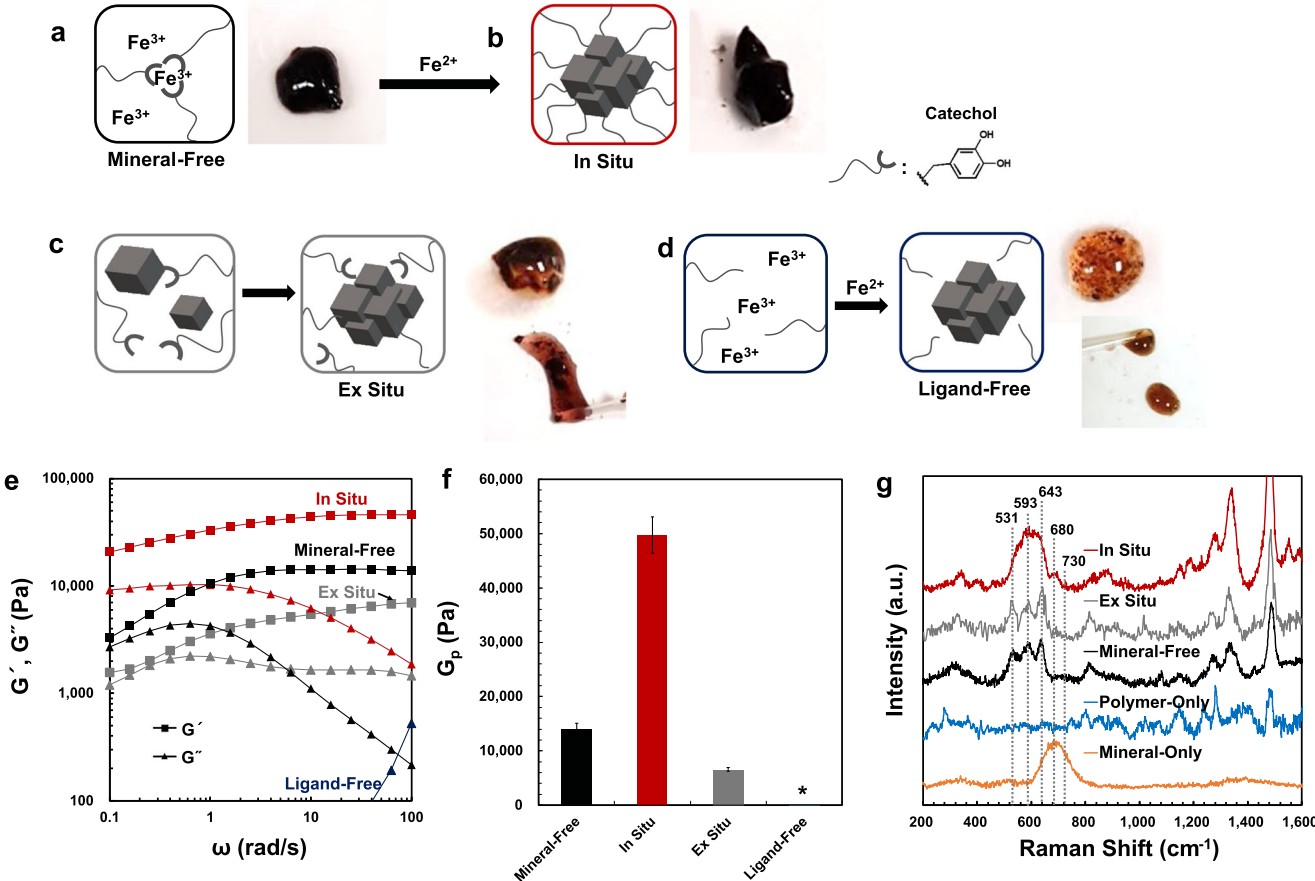

**Fig. 1 In situ crosslink mineralization via metal-coordination causes mechanical reinforcement of polymer hydrogels. a** The Mineral-Free gels were initially crosslinked via mixing a 1:1 stoichiometric ratio of $Fe^{3+}$:catechol then raising the pH to ~12 with NaOH thereby forming a network with a combination of $Fe^{3+}$-catechol coordinate crosslinks and covalent catechol crosslinks (Mineral-Free, black). **b** The In Situ gels were formed by introducing $Fe^{2+}$ to the Mineral-Free gel at a 1:3 ratio of $Fe^{2+}$:$Fe^{3+}$ to initiate in situ gel mineralization of $Fe_3O_4$ (In Situ, red). Control samples consist of gels with identical composition to In Situ gels, but where **c** $Fe_3O_4$ minerals were first grown ex situ and then mixed with 4cPEG (Ex Situ, gray), and **d** identical protocol as In Situ, but using a 4-arm PEG with no catechol modification (Ligand-Free, navy). Inset images are representative photographs of each type of sample. Note the visible mineral particle precipitation in both the ex situ and ligand-free samples. **e** Representative rheological frequency sweeps (storage modulus-$G'$, loss modulus-$G''$) and **f** plateau moduli ($G_p$) of the four different samples (**a**–**d**) at 1% strain and 25 °C. Error bars are standard deviations based on duplicate measurements from three independent samples. Note that the ex situ samples only display weak gel-like behavior and the ligand-free samples are liquids in agreement with the qualitative appearance of the samples in **c** and **d**. The asterisk indicates a missing plateau modulus value due to the lack of storage modulus ($G'$) in the ligand-free liquid samples. **g** Raman spectra of In Situ, Ex Situ, Mineral-Free as well as Polymer-Only (4cPEG alone, blue), and Mineral-Only (mineral with no polymer, orange) samples. The resonance triplet peaks characteristic of $Fe^{3+}$-catechol coordinate interactions at 531, 593, and 643 cm$^{-1}$ are broadened by in situ mineralization. The 680 and 730 peaks indicate magnetite and maghemite, respectively. Note that the schematic drawings in **a**–**d** are conceptual and not meant to describe the detailed structure of the network crosslinks nor the minerals.

730 cm$^{-1}$ indicative of $Fe_2O_3$ (maghemite)[24]. We note, however, that a mixture of iron oxide phases is likely to result from our mineralization process and continued oxidation in air. Together these findings support not only the existence of iron oxide mineral particles in the In Situ network, but more importantly a direct interaction between catechol and mineral particles in the network crosslinks. In contrast, the Raman spectrum of the Ex Situ gel suggests that the dominant polymer-metal interaction within these networks remains $Fe^{3+}$-ion-catechol coordination with little to no mineral surface coordinate interactions, in support of poor integration of pre-formed minerals within these gels.

**Additional mineralization and gel solidification.** Next, we examined the effect of additional mineralization on the properties of the In Situ hydrogel to test if repeated mineralization cycles would further grow $Fe_3O_4$ particles at coordinate crosslink sites (Fig. 2a). To synthesize this series of gel samples with increasing

levels of mineralization, we added equivalent stoichiometric amounts of $Fe^{3+}$, $Fe^{2+}$, and $OH^-$ ions to an In Situ gel for a total of four additional mineralization cycles, thereby generating a series of samples labeled In Situ ×1–×5 (please see the "Methods" for details on sample preparation). As evidenced by transmission electron microscopy (TEM) images of thin sections of In Situ ×1 and ×5 gel samples shown in Fig. 2b, c, the in situ grown mineral particles are well-dispersed in the metal-coordinate gel networks. While spherical particles of only a few nanometers in diameter dominate the In Situ ×1 gels, larger particles up to tens of nanometers in diameter (indicated with arrows in Fig. 2c) are additionally observed in the In Situ ×5 gels, alongside thin, rod-like particles present in both samples (see Table 1 in "Methods" section, Supplementary Figs. 4 and 5 for particle size distributions in In Situ ×1 and ×5, respectively). The larger particles in the In Situ ×5 gels appear to be composed of aggregates of smaller particles, and high-resolution TEM clearly displays consistent lattice fringes, suggesting that the individual particles in the

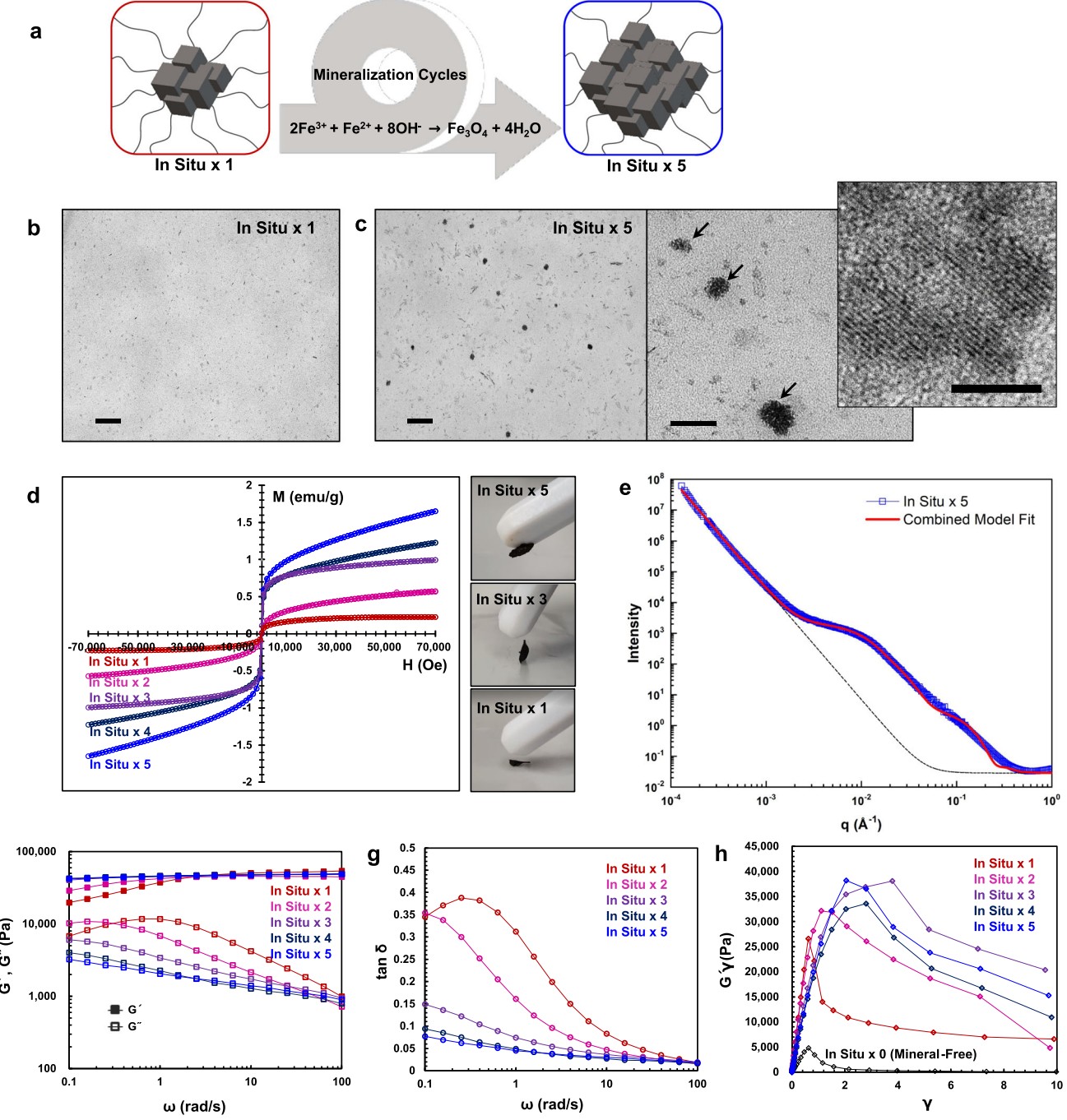

**Fig. 2 Additional mineralization causes further growth of well-dispersed mineral particles. a** In Situ gels were further mineralized by repeatedly adding equivalent amounts of mineralizing components to an already established In Situ ×1 gel scaffold through up to five mineralization cycles (labeled as In Situ ×1 through ×5, colored red, magenta, purple, navy and blue, respectively.). TEM images of **b** In Situ ×1 (scale bar is 200 nm) and **c** In Situ ×5 gels (left, scale bar is 200 nm, and right, scale bar is 50 nm), show well-dispersed particles. The inset in the In Situ ×5 panel is a representative high-resolution TEM image of one of the large aggregates of particles (marked with arrows) with clear lattice fringes indicating single-crystal structure (scale bar is 5 nm). **d** Magnetization curves and photographs of dehydrated In Situ samples, exhibiting increased magnetization and increased attraction to a magnet with increasing mineralization cycles. **e** USAXS results for the In Situ ×5 gels. The model used (red solid line) is based on image analyses of spherical and cylindrical objects identified in TEM and demonstrates a good fit with the scattering data (blue squares). The black dashed line indicates the contribution from the polymer scaffold as discussed in the main text. **f** Representative rheological frequency sweeps of In Situ gels at 1% strain and 25 °C. **g** Tan $\delta$ ($G''/G'$) curves of the same data in **f**, highlighting the evolution of increasingly mineralized gels into more solid-like rubbery materials. **h** Elastic stress–strain curves obtained from LAOS experiments on In Situ gels highlighting the mineral-induced mechanical reinforcement displayed in both ultimate strength and strain energy density.

**Table 1 Particle type and size assignment.**

| Particle type | Large spheres | Small spheres | Rods |
|---|---|---|---|
| Size (nm, pixels) | 6.2 (±2.5) nm, 200–10000 | 1.4 (±0.3) nm, 20–200 | L: 14.4 (±5.3) nm, W: 3.3 (±1.5) nm, 50–1000 |
| Circularity | 0.6–1 | 0.7–1 | 0–0.4 |

aggregate are a single crystal[29–32]. The lattice fringes shown in the inset (Fig. 2c) correspond to (220) planes of magnetite (Supplementary Fig. 6). Based on TEM area fractions, we estimate a total mineral content of $\Phi_{microscopy} \sim 0.29$ vol% in the hydrated In Situ ×5 gel, which is in good agreement with the predicted maximum when assuming complete mineralization of $\Phi_{maximum} \sim 0.37$ vol%, especially taking into account the small, but unavoidable volume change of the gel during TEM specimen preparation (please see "Methods" for details). All of the mineralized gels show superparamagnetic behavior typical of $Fe_3O_4$ nanoparticles[33,34], and as the volume fraction of mineral increases with mineralization, the magnetic properties of the gels are enhanced, as shown in Fig. 2d. This behavior offers a magnetization-based estimate of the mineral volume fraction in each hydrogel. Specifically, by comparing the maximum magnetization of each sample with the saturation magnetization for pure bulk $Fe_3O_4$ of 86 emu/g[35], we estimate a mineral volume fraction of $\Phi_{magnetic} \sim 0.28$ vol% for the In Situ ×5 hydrogel, which closely corresponds to the $\Phi_{microscopy} \sim 0.29$ vol% estimated from the above TEM image analysis (Supplementary Figs. 7 and 8). Furthermore, $\Phi_{magnetic}$ increases linearly across all five reaction cycles indicating well-controlled in situ gel mineralization. USAXS analyses of the In Situ ×5 gels (Fig. 2e) provide complementary evidence for size and spatial distribution of the mineral content in the In Situ gels as observed in TEM (Supplementary Figs. 9 and 10, and Supplementary Table 1 for model parameters and additional details on the analysis). Importantly, the scattering data is well-captured by a model that combines scattering contributions from particle size and distribution parameters identified from TEM (Supplementary Fig. 9). A monotonic scattering at low $q$ is also observed (Fig. 2e black dashed line), indicating spatial features with a scale that clearly exceeds the 1 µm length-scale limit of USAXS; such large features are absent in TEM images, and thus we attribute this to the macrostructure of the gel as commonly observed in other studies[36,37]. In contrast to the In Situ gel, the Mineral-Free gel exhibits substantially weaker scattering in the higher $q$ regime and no discernable scattering below $q = 0.1$ Å as expected (Supplementary Fig. 10). Overall, the absence of structure factor correlations at high $q$ and larger scattering objects at low $q$ in the USAXS data support the notion of small well-dispersed minerals in our In Situ gels.

Figure 2f displays representative SAOS frequency sweeps of the series of In Situ ×1 to ×5 gels, demonstrating a progressive transformation into gels with increasingly rubbery solid-like behavior with further mineralization as evidenced by the increasing storage modulus ($G'$) at low frequencies (long timescales) and overall decreasing loss modulus ($G''$). This mechanical evidence of network solidification through crosslink mineralization is more clearly shown in Fig. 2g, which illustrates the consistent stepwise decrease in tan $\delta$ (tan $\delta = G''/G'$), and therefore total network dissipation, with each cycle of mineralization. Similarly, the characteristic time scale required for the bulk hydrogel networks to relax stress following a step-strain test also increases with each mineralization cycle in accordance with more solid-like behavior (Supplementary Fig. 11). To examine how increasing mineralization changes gel properties during

deformation to failure, we used large amplitude oscillatory shear (LAOS) rheology (Fig. 2h, Supplementary Fig. 12), as well as tensile tests (please see Supplementary Figs. 13 and 14 for details). We show the evolution of $G'\gamma$ (i.e., the elastic component of the total sample stress) as a function of strain amplitude $\gamma$ in Fig. 2h, which represents a rheological analog of a tensile stress–strain plot[38]. The figure reveals a significant increase in the ultimate shear strength (i.e., the maximum in $G'\gamma$) upon mineralization. This mineral-induced increase in ultimate strength is also observed in tensile tests (Supplementary Figs. 13 and 14). Furthermore, the mineralized gels exhibit increased energy dissipation during failure (demonstrated by a maximum in $G''$ in LAOS tests, Supplementary Fig. 12)—a feature absent in the Mineral-Free systems—which suggests that mineralization introduces a discrete energy dissipation mode in the gel plausibly via mineral-catechol interfacial bond-breakage analogous to the filler-induced Payne effect.

**Maximizing gel stiffness with minimal mineralization and its proposed mechanism.** Overall, our data suggest that in situ mineralization can reinforce the gel network using only a small amount of minerals in a remarkably efficient manner. This becomes clear when the mineral-induced change in stiffness of In Situ gels is compared to that of previously reported conventional $Fe_3O_4$ nanoparticle gels composed of the same polymer assembled with pre-synthesized nanoparticles of similar size[15] (Fig. 3 and Supplementary Table 2). As illustrated in Fig. 3, when compared to a corresponding nanoparticle-free metal-ion crosslinked gel, the gel stiffness increases by ~35,000 Pa upon in situ network mineralization with only ~0.2 vol% nanoparticles. In contrast, despite a tenfold higher content of $Fe_3O_4$ nanoparticles (~2 vol%), a similar analysis of a conventional nanoparticle gel shows a decrease of ~5000 Pa in gel stiffness, a common effect explained by the introduction of elastically inactive polymer chains such as loops, and its associated decrease in network elasticity, upon conventional mixing in of pre-synthesized nanoparticles in gels[15,19–21]. Our findings thereby support the hypothesis that mineralizing even a small amount of $Fe_3O_4$ particles directly at the gel network crosslink sites leads to significant improvement in gel reinforcement, in contrast to incorporating pre-existing particles into the polymer network, which can actually decrease gel stiffness. Furthermore, our data suggests that the in situ nucleation and growth of minerals directly at the gel network crosslink sites not only suppresses the traditional introduction of topological defects observed upon conventional mineral particle incorporation, but might even trigger a mechanism of mineral-induced recruitment of elastically inactive polymer network chains. Specifically, the observation that only the first cycle of mineralization causes a significant increase in gel modulus could plausibly be explained by the transformation of initial low functionality bis- or tris-catechol-$Fe^{3+}$ coordinate complexes into high functionality catechol-mineral crosslink structures through the recruitment of polymer network chains initially elastically inactive in the Mineral-Free gel during the process of nucleation and growth of particles directly in the metal-coordinate crosslinks (Supplementary Fig. 15 for more in depth discussion on this mechanism)[39,40]. Furthermore, while we cannot rule out possible particle nucleation outside the coordinate crosslink sites, our Raman (Supplementary Fig. 16) and mechanical data (Fig. 3 and Supplementary Figs. 17 and 18) suggest that later mineralization cycles result in further growth of the particles already formed, and it is tempting to speculate if mineral-bound catechol ligands become entrapped in growing particles[41] during these additional cycles of mineralization. Such ligand-mineral entrapment could potentially both explain the lack

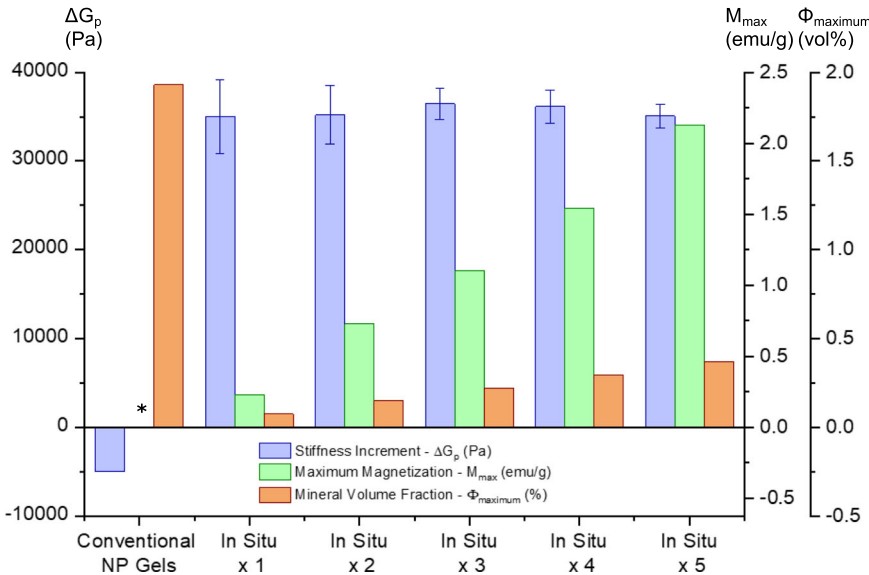

**Fig. 3 In situ mineralization increases gel stiffness with minimal use of minerals.** The change in the stiffness ($\Delta G_p$, left axis, blue bars, the difference between the moduli of mineral-containing gels and their corresponding mineral-free metal-ion crosslinked gels), maximum magnetization ($M_{max}$, middle axis, green bars, from Supplementary Fig. 7) and the mineral volume fraction ($\Phi_{maximum}$, right axis, orange bars) of conventional $Fe_3O_4$ nanoparticle gels (from ref. [15]) and In Situ gels (this work). The gel stiffness increases by ~35,000 Pa upon in situ incorporation of only ~0.2 vol% minerals but decreases by ~5000 Pa upon traditional incorporation of ~2 vol% pre-synthesized nanoparticles. The $M_{max}$ increases linearly with each in situ mineralization cycle without compromising stiffness. The asterisk indicates that the gel magnetic properties were not quantified in this past work[15]. The error bars indicate standard deviations from three independent samples.

of increased moduli and the slower stress-relaxation of more mineralized gels, as well as the diminishing Raman resonance for $Fe^{3+}$-catechol coordination (~530 to ~650 $cm^{-1}$) and increase in the peak for iron oxides (~680 $cm^{-1}$) observed after five cycles of mineralization (Supplementary Fig. 16). Although our conjecture does not allow a detailed explanation of how the ion- to particle-coordinated crosslink transformation takes place, an additional benefit to increasing gel nanoparticle content via direct crosslink mineralization is that, in contrast to magnetic hydrogel systems with mechanically dispersed particulate phases, this in situ method allows us to avoid the typical tradeoff between magnetic performance and mechanical properties of the resulting gels (as illustrated in Fig. 3)[21].

**In situ mineralization of other metal-ligand systems.** Finally, to demonstrate the generality of our in situ mineralization approach, we synthesized gels with different metal-ligand coordination complex crosslinks (Fig. 4 and Supplementary Figs. 19–22). We used the same 4-arm PEG polymer backbone but functionalized with histidine ligands (4hPEG) instead of catechol, and first prepared Mineral-Free gels with metal-coordinate crosslinks with either $Ni^{2+}$ or $Cu^{2+}$ ions[42]. Importantly, unlike the oxidation-prone catechol-based gel networks above, these histidine-metal-coordinate networks do not contain covalent crosslinks, hence they behave close to purely transient viscoelastic fluids before any network mineralization, as has been reported in previous work[42–44]. We next introduced additional $Ni^{2+}$ and $Cu^{2+}$ ions into these fluid networks to induce in situ mineralization of $Ni(OH)_2$ and $Cu(OH)_2$, respectively (more information on mineral morphologies and compositions available in Supplementary Figs. 20–22). Both Ni-(Fig. 4a, b) and Cu-histidine networks (Fig. 4c, d) underwent drastic changes in mechanical properties with mineralization, both becoming solids compared to their Mineral-Free counterparts, as evidenced by the changes in the storage ($G'$) and loss modulus ($G''$) such that $G' > G''$ at all measured frequencies and slower stress-relaxation (Supplementary Fig. 19). The dramatic switch to overall

solid-like rheological behavior, slow-down of the stress-relaxation and increase in $G_p$ upon mineralization of purely transient viscoelastic fluids to our knowledge have not been demonstrated before. As observed in the Fe-catechol system, analogous Ex Situ gels showed a loss in stiffness plausibly due to inefficient crosslinking from mixing in pre-existing minerals, while no gelation occurred without metal-coordinating ligands on the polymer backbone (Ligand-Free). We also note the possible inclusion of impurities other than metal hydroxides, however, regardless of the mineral identity, our results demonstrate the generality of the in situ mineralization-induced reinforcement of metal-coordinate polymer networks.

## Discussion

Mineralization in macromolecular hydrogel networks is a broad field of study, which have focused on various important topics such as mineral morphogenesis control[45–49], mineral incorporation of macromolecules[41,50,51], as well as the influence of mineralization on mechanical properties, for example of solid nacre-like nanocomposites[52,53]. Here, we have shown that metal-coordinate crosslinked macromolecular hydrogel networks appear to offer new opportunities to direct the growth of nanoparticles through in situ gel mineralization. By nucleating minerals directly at the coordinate complex crosslinking sites, we were able to better control mineralization spatially, while mechanically reinforcing the hydrogel network using only a small amount of minerals and minimizing network defect formation typically associated with a more conventional incorporation of particles such as mixing. This approach was shown to be generally applicable to different types of metal-coordinate crosslinking systems. Furthermore, through repeated mineralization cycles, we demonstrated the ability to increase mineral content and hydrogel magnetization without sacrificing gel stiffness and strength. We thereby achieved a significant increase in stiffness using only a small amount of minerals compared with conventional nanoparticle composite gels. We note that the scope of this paper was not to mimic hard-condensed biological composite

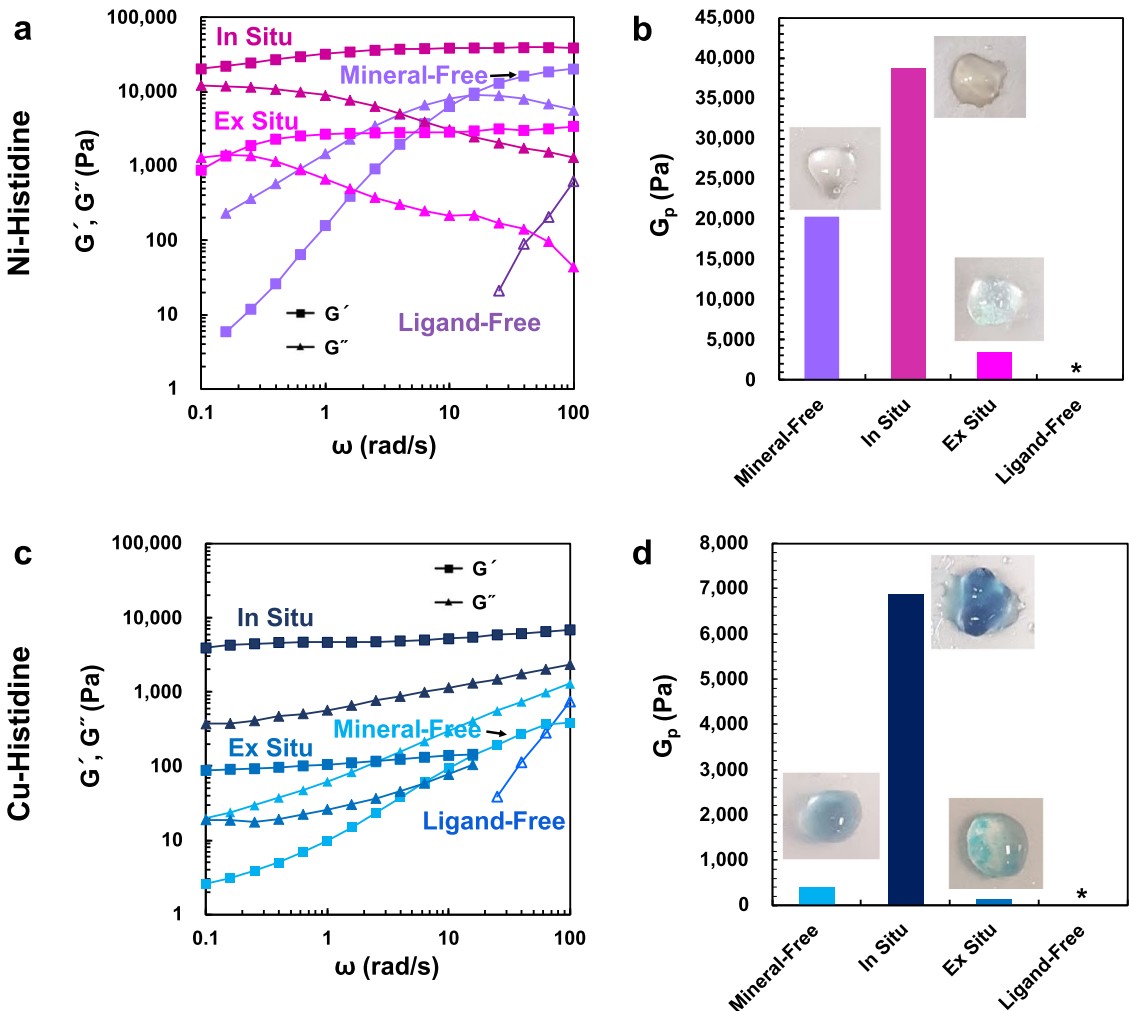

**Fig. 4 Mechanical reinforcement via in situ mineralization in other metal-coordinate crosslinked networks.** Using noncovalent metal-coordinate networks crosslinked by Ni- or Cu-histidine-coordination, the same four processing pathways (i.e., Mineral-Free—blue violet, skyblue; In Situ—red violet, navy; Ex Situ—magenta, sapphire blue; Ligand-Free—purple, blue, respectively for Ni- and Cu-systems) as in Fig. 1 were tested. The Mineral-Free samples were formed via mixing the metal ions with 4hPEG in 1:2 stoichiometric ratios of metal ions:histidine, then raising pH to ~12 using NaOH. The in situ samples were prepared by inducing $Ni(OH)_2$ or $Cu(OH)_2$ nucleation via additional introduction of $Ni^{2+}$ and $Cu^{2+}$ ions to the Mineral-Free systems through reaction with the excess $OH^-$ (final metal ions:histidine = 1:1). **a** Linear viscoelastic frequency sweeps (storage modulus-$G'$, loss modulus-$G''$) and **b** plateau moduli ($G_p$) of the samples from the four different processing pathways of the Ni-histidine system. **c** Linear viscoelastic frequency sweeps and **d** $G_p$ of the Cu-histidine system. Inset images are representative photographs of each type of sample. Note that precipitation of particles is visible in the Ex Situ gels. The rheological measurements were performed at 1% strain and 25 °C. The asterisk indicates missing values when $G'$ is not detectable.

materials such as nacre[52,53] or chiton teeth[3,54]. Rather we focused on a fundamental exploration of a new bioinspired approach to reinforcement of organic–inorganic soft-condensed matter that could prove advantageous and efficient compared to conventional routes. Nonetheless, our work suggests that we can simultaneously reinforce and functionalize hydrogels with magnetic properties for potential use in soft actuation, drug delivery, and tissue engineering applications[6–9,55,56]. More broadly, the bioinspired approach to material solidification through targeted crosslink mineralization of aqueous metal-coordinate polymer networks presented here, could offer both resource- and cost-efficient processing pathways required to meet the future demands of sustainable manufacturing of organic–inorganic composite materials.

## Methods

**Materials.** Ten kilodaltons 4-arm-PEG-NHS (4aPEG-NHS) and 10 kDa 4-arm-PEG-NH₂ (4aPEG-NH₂) were purchased from JenKem Technology USA, Inc. Inorganic salts including iron(III) chloride hexahydrate (FeCl₃ · 6H₂O), iron(II)

chloride tetrahydrate (FeCl₂ · 4H₂O), nickel(II) chloride hexahydrate (NiCl₂ · 6H₂O), copper(II) chloride hexahydrate (CuCl₂ · 6H₂O), calcium chloride (CaCl₂), and sodium hydroxide (NaOH) were purchased from Sigma-Aldrich unless indicated otherwise. All other organic ingredients were purchased from Sigma-Aldrich.

**Synthesis of 4cPEG.** 4-arm-PEG-catechols (4cPEG) was synthesized following a previously reported process[44] with some modifications: dopamine hydrochloride (dopa-HCl) (1.5× molar relative to –NHS) was mixed with 4aPEG-NHS and dissolved in the 10 mL of anhydrous N,N-dimethylformamide (DMF) in the round-bottom flask. The reaction was initiated by adding triethylamine (TEA) (2.5× molar equivalent relative to –NHS) to substitute –NHS to –catechols in the flask protected with N₂ gas. The reaction was proceeded for 12 h with magnetic stirring in the flask in the 55 °C silicone oil bath to produce 4cPEG. Then, the flask was cooled down to room temperature. Then, organic phase with the product was separated from the aqueous phase with the salts via adding 20 mL of chloroform and 20 mL of water. Subsequently, the organic phase was extracted through separatory funnel. The remnant water from the organic solution was removed by adding the Na₂SO₄ to aggregate with water and precipitate. Then the organic solvent was vaporized using a rotary evaporator. The product was further purified via precipitating in diethylether, followed by re-dissolving in 5 mL of dichloromethane (DCM), then re-precipitating in diethylether. After repeating the above

purification steps 3+ times, the precipitate was dried via vacuum, then freeze-dried to obtain the final product (4cPEG).

**Synthesis of 4hPEG**. 4-arm-PEG-histidine (4hPEG) was synthesized following the previously reported method[42]. In brief, 5.0 g of 4aPEG-NH$_2$ (10 kDa), 1.49 g of Boc-His(Trt)-OH and 1.33 g of BOP reagent were mixed in the 15 mL of dichloromethane (DCM) in the Schlenk flask protected with N$_2$ gas. 1.07 mL of N,N-diisopropylethylamine (DIPEA) was then added to initiate the reaction and proceeded for 2 h. The product was precipitated in diethylether, followed by purification through dissolving and precipitating in methanol under −20 °C, 3+ times, then re-precipitated in diethylether. The product was dried under vacuum, then went through the deprotection reaction for 2 h in the cleavage solution comprising 95 mL trifluoroacetic acid (TFA), 2.5 mL of triisopropylsilane and 2.5 mL of water. The solvent was then vaporized via rotary evaporation and the product was purified again through precipitating in diethylether and methanol 3+ times as described before. The final product (4hPEG) was obtained by freeze-drying as described above for 4cPEG.

**Formation of Fe-4cPEG hydrogels**. For hydrogels without mineralization, an [Fe$^{3+}$]:[catechol] ratio of 3:3 ("3:3" gels) and 1:3 ("1:3" gels) were used by mixing an appropriate volume of 0.2 M FeCl$_3$·6H$_2$O (aq) with a 200 mg/mL 4cPEG solution (12.5 μL), then raising pH up to ~12 via mixing 4 M NaOH (aq) to produce 20 μL hydrogels of 125 mg/mL polymer concentration. The final concentrations of Fe$^{3+}$ were 50 mM for the 3:3 gels and 16.67 mM for the 1:3 gels. The final concentrations of NaOH were 180 mM for the 3:3 gels and 100 mM for the 1:3 gels. Unless specified otherwise, the 3:3 gels are the reference sample without mineralization, which is labeled as "Mineral-Free" or "In Situ × 0" gels according to context.

**Formation of Fe-mineral-4cPEG hydrogels**. The Fe-mineral-4cPEG hydrogels (20 μL) were formed following two different methods: (1) minerals were grown within the hydrogel network in situ (labeled as "In Situ") and (2) minerals were grown separate first, then mixed with polymer for gel formation (labeled as "Ex Situ") reminiscent of the formation process of conventional Fe$_3$O$_4$-nanoparticle gels[15]. The In Situ gels were formed via first following the same steps for forming the Mineral-Free (3:3) gel (18.33 μL) described above, then carefully mixing 0.2 M FeCl$_2$·4H$_2$O (1.67 μL) to trigger in situ mineralization in the hydrogel (20 μL). The mineralization reaction inside the hydrogel is designed to achieve stoichiometric ratios of [Fe$^{2+}$]:[Fe$^{3+}$]:[catechol] of 1:3:3 to match with the Fe$_3$O$_4$ mineralization reaction:

$$Fe^{2+} + 2Fe^{3+} + 8OH^- \rightarrow Fe_3O_4 + 4H_2O$$

whereby the added 1× stoichiometric amount of Fe$^{2+}$ is expected to react with the 2× stoichiometric amount of Fe$^{3+}$ to form a 1× stoichiometric amount of Fe$_3$O$_4$[57,58], while the leftover 1× stoichiometric amount of Fe$^{3+}$ can participate in tris-catechol-Fe$^{3+}$ coordination complexation, which dominates at pH ~12. Note that the sample was left for 5 min to equilibrate before rheological measurement. Thus, this basic protocol should hypothetically produce one Fe$_3$O$_4$ per metal-coordinate crosslinking site naively assuming 100% yield. In contrast, the Ex Situ gels were formed by separately mixing the same exact composition of inorganic ingredients (FeCl$_3$·6H$_2$O, FeCl$_2$·4H$_2$O, NaOH) as the In Situ gels to induce mineralization, thus forming a dispersion with precipitates first (Mineral-Only), before mixing with 4cPEG (200 mg/mL) to form the hydrogel. In both gels, the final concentrations of Fe$^{3+}$, Fe$^{2+}$, and NaOH were 50 mM, 16.67 mM, and 190 mM, respectively. For additional comparison, the 1:3 gels were also subjected to the mineralization protocol described above, yet in this case the [Fe$^{2+}$]:[Fe$^{3+}$]:[catechol] ratio was adjusted to 0.5:1:3 with the final concentration of [Fe$^{2+}$] in the hydrogel equal to 8.33 mM. Here, the naive assumption is made that all added Fe$^{2+}$ will react with only Fe$^{3+}$ ions already participating in tris-catechol coordination complexation. The In Situ gels exposed to additional cycles of mineralization (labeled as "In Situ × n", n: 2, 3, 4, 5) were formed by first producing an In Situ gel as above (labeled as "In Situ × 1" in such context), dehydrating this gel for 24 h (25 °C, 16 % R.H.), removing excess ions by carefully rinsing and dabbing followed by another 24 h of dehydration, and then following the same sequence of adding components at identical concentrations as when producing the In Situ × 1 gel (i.e., adding polymer-free water, FeCl$_3$ (aq), NaOH (aq), then FeCl$_2$ (aq)), overall thereby recovering the original volume (20 μL) of the hydrogel. Thus-formed gels are labeled as "In Situ × 2" as they went through the mineralization process twice. In this manner, the In Situ × n gels were produced by repeating this same mineralization cycle up to five times.

**Formation of Ni- or Cu-4hPEG hydrogels**. Ni- or Cu-4hPEG hydrogels (20 μL) were formed via adding the 0.2 M NiCl$_2$·6H$_2$O or CuCl$_2$·6H$_2$O to the 200 mg/mL 4cPEG aqueous solution (12.5 μL) followed by pH jump by NaOH. The hydrogel was formed in the [Ni$^{2+}$ or Cu$^{2+}$]:[histidine] ratio of 1:2, then the pH was controlled to ~12 via adding NaOH to form bis-metal-histidine coordinate gels. The final polymer concentration in hydrogels was 125 mg/mL. The final concentration of Ni$^{2+}$ or Cu$^{2+}$ was 25 mM and that of NaOH was 125 mM.

**Formation of Ni- or Cu-mineral-4hPEG hydrogels**. The Ni- or Cu-mineral-4hPEG hydrogels (20 μL) were formed via first following the regular Ni- or Cu-4hPEG hydrogels (17.5 μL) processing as described above, then adding 0.2 M aqueous solution of the equal amount of Ni or Cu ions (2.5 μL, instead of the pure water in the mineral-free histidine gel) right after the pH jump to induce the following reaction while keeping the pH level for maintaining metal-histidine coordination:

$$Ni^{2+} + 2OH^- \rightarrow Ni(OH)_2$$

$$Cu^{2+} + 2OH^- \rightarrow Cu(OH)_2$$

Thus, the final concentration of Ni$^{2+}$ or Cu$^{2+}$ in the hydrogel was 50 mM and that of NaOH was 125 mM.

**Rheological measurements**. The rheological measurements were performed using an Anton Paar MCR 302 rheometer (Anton Paar). A hydrogel specimen (20 μL) was placed under the 10 mm-diameter parallel plate geometry (PP10). The measurement was conducted at 25 °C with a solvent trap for retarding evaporation during the process. Oscillatory frequency sweeps were performed at constant 1% strain and angular frequency (ω) ranging from 100 to 0.1 rad/s to record the storage (G′) and loss (G″) moduli values, and corresponding phase angle (tan δ = G″/G′). Stress-relaxation tests were performed under 10% step strain (γ$_0$), and the shear stress (σ) and relaxation modulus (G(t) = σ(t)/γ$_0$) were recorded over 1000 s. The relaxation modulus-G(t) was normalized by G$_i$ (the modulus at time 0.1 s when the strain was stabilized at the set value). Strain-sweep tests were performed under oscillating strain amplitudes ranging from 0.001 to 10 (i.e., 0.001 ≤ γ$_0$ ≤ 10) at a constant frequency of ω = 10 rad/s. No visual rupture or edge fracture of the specimen was observed after the measurement.

**Tensile test**. The tensile experiments were performed using a CellScale UStretch mechanical tester with a 0.5 N load-cell (CellScale Biomaterials Testing). Gel specimens (60–80 μL) were prepared in dog-bone shaped molds (gauge length 4.0 mm, width 1.9 mm and depth 1.4 mm) at 50% of the overall concentration of the hydrogels described for rheological measurements for reliable controllability in practical sample preparation. Each end of the specimen was bonded to two acrylic sheets by Scotch Superglue (3M), then the acrylic sheets were loaded on the grips to prevent any damage to the hydrogel by the direct contact with the grip. All experiments are performed at a displacement speed of 0.83 mm/s (i.e., a strain rate at $\bar{\varepsilon} = 0.208\,s^{-1}$ for the 4.0 mm measured length of gel specimens). The tensile stress (σ) and strain (ε) were calculated from the actual sample dimensions measured before each run.

**Raman spectroscopy and microscopy**. Raman spectroscopic analysis was performed using an HR800 Raman Spectrometer—LabRAM Raman confocal microscope (HORIBA Jobin Yvon) with 785 nm near-IR laser excitation. The dry specimen (dehydrated for 48 h at 20 °C, 16% R.H.) was loaded on a glass substrate, then placed on a Märzhäuser stage (Märzhäuser Wetzlar) positioned under the 10× microscope lens. The laser power was 30 mW, the grating was 600 g/mm and the filter was adjusted to 10%. The integration time was 2 s with an accumulation of five times. Raman spectra and microscopic images were obtained using LabSpec 6 (Horiba Scientific) software.

**Transmission electron microscopy (TEM)**. To prepare samples for TEM, hydrated gels (20 μL) were placed into 100% ethanol overnight and then infused with EMBED-812, first with a 50/50 ethanol/EMBED-812 mixture overnight and then with 100% EMBED-812 overnight. The EMBED-812 was then polymerized. We note that while some sample volume change caused by the solvent exchange process is expected, the sample was always kept in a swollen state with respective solvents and any observed volume change was not significant. Sections were cut on a Leica EM UC7 ultramicrotome with a Diatome diamond knife at a thickness setting of 50 nm. The sections were examined using an FEI Tecnai spirit at 80 kV and photographed with an AMT CCD camera. The images were used for particle size and volume fraction analyses. A JEOL JEM-2010 high-resolution TEM operating at 200 kV was used to take higher magnification images of the iron oxide particles. The large spherical particles showed clear lattice fringes, indicating single-crystal structure.

**TEM image analysis**. To obtain the average sizes, size distributions, and volume fractions of grown iron oxide particles, we processed the low-magnification TEM images using ImageJ software. To process the images for particle counting, we used the image contrast to separate the particles from the background using the "RenyiEntropy" auto threshold method. We then cleaned up the image and counted and sized the particles. We observed and classified three different types of particles: "Small" sphere, "Large" spheres, and "Rods". We used different constraints to count each type of particle, shown in the Table 1 below. We also analyzed larger "Aggregates" found in some images (23.4 ± 12 nm). Owing to the small number of Aggregates, the analysis was done manually.

The lattice fringes (d-spacing) of "Aggregates" particles were also measured to characterize crystal identity using ImageJ from the high-magnification TEM images.

**Small-angle scattering (SAXS)**. Data were collected using Ultra-Small-Angle and Small-Angle X-ray Scattering (USAXS/SAXS) instrument at 9-ID beamline at the Advanced Photon Source (APS). Instrument design and details are described elsewhere[59]. Hydrated samples (20 μL) were mounted in cells with sample thickness of 1 mm and with 3M Scotch tape on either side. X-ray energy was 21 keV (wavelength = 0.5905 Å), beam size was 0.6 mm × 0.6 mm for USAXS and 0.1 mm × 0.6 mm for SAXS. Data collection was 90 s for USAXS and 15 s for SAXS. Data were subtracted from a blank cell with scotch tapes, reduced, and then merged together using software (*Indra*, *Nika*, and *Irena*) provided by the beamline[59–61]. Slit smearing inherent to the USAXS instrument was removed using de-smearing routines in *Irena* to generate pinhole equivalent data for analysis. Scattering models as described in the text were fitted using Irena package[61] (Supplementary Information for details).

**Magnetization analysis**. Magnetization measurements were carried out using a Magnetic Property Measurement Systems 3 (MPMS3) Superconducting Quantum Interference Device (SQUID) Vibrating-Sample Magnetometer (VSM) (Quantum Design). The dry sample (dehydrated in the same condition used for preparing Raman specimen) was mounted in the clear polypropylene straw sample holder. After loading the sample into the apparatus, the magnetization (M) loop measurement was proceeded using the DC scanning mode from an applied magnetic field (H) ranging from −70,000 to 70,000 Oe at 300 K.

**Scanning electron microscopy (SEM)**. The grown minerals or freeze-dried gels were characterized using a Zeiss Merlin High-resolution SEM at 1−3 kV accelerating voltage and 100–300 pA beam current, using depth of field or high-resolution (HE-SE2) mode, to determine their size and morphology.

**Energy dispersive spectroscopy (EDS)**. The elemental analysis on the specimen was performed using the EDS accessory equipped on the aforementioned Zeiss Merlin High-resolution SEM. The measurement was proceeded using Analytic column mode, 5 kV accelerating voltage and 200 pA beam current. The input count per second was ~2000 cps. The minerals or polymer matrix were examined using point analysis.

**X-ray diffraction analysis (XRD)**. The XRD analysis on gel specimen was performed using Bruker D8 General Area Detector Diffraction System (GADDS) Multipurpose Diffractometer. The dehydrated gel (as used in Raman spectroscopy) was placed on Eulerian ¼ cradle and the diffraction data were collected over a 2θ range 15° to 55° using Cu Kα radiation at 40 kV, 40 mA.

## Data availability

The datasets generated during and/or analyzed during the current study are available from the corresponding author on reasonable request.

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

## Acknowledgements

S.K. acknowledges the financial support from the 12th Samsung Scholarship Program. J.S., G.H.M., and N.H.-A. acknowledge funding from the U.S. Army Research Office through the Institute for Soldier Nanotechnologies at MIT, under Contract Number W911NF-13-D-0001. TEM work was conducted utilizing the W.M. Keck Foundation Biological Imaging Facility at the Whitehead Institute with the aid of Nicki Watson. This research used resources of the Advanced Photon Source, a U.S. Department of Energy (DOE) Office of Science User Facility operated for the DOE Office of Science by Argonne National Laboratory under Contract No. DE-AC02-06CH11357. This work was supported in part by the MRSEC Program of the National Science Foundation under Award DMR-1419807. See USAXS web site for more details: https://usaxs.xray.aps.anl.gov/, contact: usaxs@aps.anl.gov

## Author contributions

S.K., A.U.R., and N.H.-A. conceived and planned the project. S.K and A.U.R. coordinated to conduct the experiments and analyze data. S.K. designed the experiments and produced the figures. A.U.R. performed HRTEM image analysis. J.S. and J.I. performed USAXS characterization and analysis. S.K., A.U.R., J.S., and N.H.-A. prepared the manuscript. G.H.M. provided discussion. All authors discussed the results and commented on the manuscript.

## Competing interests

The authors declare no competing interests.
