## [Peer Review File · Nature Communications]

Reviewers' comments:

Reviewer #1 (Remarks to the Author):

Holten-Andersen and coauthors demonstrate that metal-ion coordination complexes can serve as nucleation sites for in situ mineralization of hydrogels, leading to mechanical property enhancement as compared to mineral-free gels, or gels that are formed with pre-assembled mineral particles. Overall, I am impressed with the approach, experimental analysis and interpretation of results. Many groups (including that of Holten-Andersen) have studied aspects of metal-ion coordinated bonding and its effects on rheology in prior work; however, this new study shows something quite different: the ability to introduce mineralization and demonstrate its impressive impacts on mechanics, which is both innovative and exciting. I believe this will be an important contribution to the biomaterials literature. That said, there are a number of improvements that I feel would strengthen the paper, as described below:

1. The emphasis in the abstract on sustainable manufacturing seems oversold, given the current set of results which are focused more on scientific analysis than process optimization and scalability. I would reframe the study in terms of the fundamental discovery aims rather than translation to manufacturing. A brief comment on future uses in the discussion may be warranted, but the current abstract is misleading, given the results and discussion that follow. If authors prefer to keep the abstract as-written, then more details of how this manufacturing method compares to current state of the art, in terms of yield and resultant material properties is warranted, as well as a discussion of the practicality of using this for true manufacturing purposes.

2. Similarly, the introduction emphasizes magnetic mineralization using iron-containing complexes, but the authors later show that it is possible to nucleate $\text{Ni}(\text{OH})_2$ and $\text{Cu}(\text{OH})_2$ as well. What is the benefit of emphasizing the iron-containing clusters (particularly given the oxidation issues associated with catecholic bonding sites)? Is this simply a good method for characterization of the sample via magnetic fields, or are there other specific advantages? This should be clarified in the revision.

3. In Figure 1, the reaction path shows $2 \text{Fe}^{3+} + 1 \text{Fe}^{2+} + 8 \text{OH}^-$, but in the caption, authors list a ratio of $\text{Fe}^{2+}:\text{Fe}^{3+}$ as 1:3. Is the Fe^{3+} added in excess? Please add more details describing how the relative ratio of ions (and other reagents) are chosen, and what (if any) effect excess agents would have.

4. Is the schematic shown in Figure 1a showing an accurate ratio of covalent and metal-coordinate crosslinks, and is the distance between the various netpoints accurate?

5. Please add error bars to the plateau modulus data shown in Figure 1f. Is the difference observed between the mineral-free and ex situ data significant? In the discussion of these data on page 5 (lines 19-21), authors refer to Figure 1c to demonstrate precipitation in the ex situ case, but this figure shows only a schematic. Are there photographs showing visible precipitation that can be included instead or added to the SI?

6. In Figure 1g, Can the relative proportion of mineralized and complexed ions be estimated from the two peaks observed in the in situ sample?

7. In Figure 2d, the in situ x4 and x5 data sets do not show clear saturation at the maximal applied H-fields. Can authors increase $|H|$ to better determine the maximal value? In Figure S6, authors indicate that they can extract a maximal value from a linear extrapolation of $1/H$ versus M but do not show this data or include the full equation to which the data are fit. These should be included. Also, why at low cycle numbers (x2, x3) do the volume fractions determined via M_s value and measured maximal value not agree, given the obvious saturation in the raw data at high $|H|$?

8. Did authors measure the magnetic properties of the ex situ mineralized gels? If so, what properties did they show? If not, why not?

9. For the tensile testing data shown in Figure S11b – how many samples were tested for each condition? Are the error bars showing standard deviation from replicate measurements? What was the hydration state of the gels during tensile testing? If different than the hydration state of the gels during LAOS, how would you expect these changes in wetting to influence the results?

10. For the LAOS data shown in Figure 2h, is there a meaning to the shear strain at which the peak value in strength occurs? Also, the caption indicates that the strain energy density increases with mineralization – how is this determined from the plots shown in Panel 2h?

11. In Figure 3, authors show the change in plateau modulus +/- mineral addition. Can authors please include the actual values or G_p before and after addition in the SI? Please also include error bars on the plot in Figure 3, propagating the uncertainty properly for the subtraction operation. Why is no magnetization data shown for the conventional NP gels?

12. Authors claim that the increased topological defects in the conventional NP case undermine the mechanical reinforcement effects these should have. Presumably the likelihood of defect formation increases with NP density. Do NP-seeded networks at volume fractions similar to those obtained with the in situ gels show better mechanical enhancement? How does the response of these gels compare to that of the in situ gels under similar NP volume fractions?

13. I don't understand the data in Figure S15. The schematic suggests that there are no catechol groups at the polymer ends – is this correct? If so, what is attracting the Fe^{3+} to the polymer end tri-junction? The caption suggests that there are catechols present but that the Fe^{3+} is added at low concentration to form nominally monovalent stoichiometries. Can you please revise either the text or schematic to clarify? Also, is the last sentence of the caption "...regardless of the presence of catecholic covalent crosslinks..." correct? Does this mean that catechols are not required for the mineralization or stiffening effect? Something seems inconsistent here.

14. The plateau modulus data in Figure 4 suggests that the ion identity has a large effect on the mechanical behavior. Can the authors please add either a brief explanation or appropriate references to other work to explain this effect? In general, is the fold increase between the mineral-free and in situ data meaningful (i.e., can this be reconciled with the single-ion binding affinity and/or and local concentration of mineral NP in the gel)?

Reviewer #2 (Remarks to the Author):

The manuscript by Kim et al. describes how significant mechanical reinforcement of hydrogels can be achieved by in situ metal coordinated crosslink mineralization. This work describes a series of carefully performed experiments that characterize the mineralized hydrogels (and controls) made of PEG end-functionalized with metal-coordinating ligands of either catechol or histidine. The authors propose that the nucleation of the mineral nanoparticles happens in the crosslinks via metal coordination, which yields to significant reinforcement. In my opinion, this paper deserves publication in Nature Communication and congratulate the authors for the beautiful results. However, there are some parts of the manuscript that require clarification before publication, which, I wish, will further strengthen this manuscript and its implications. I recognize that the manuscript contains a large amount of experimental data, and that many of them need to be in the SI. As a result, reading the manuscript is not a fluent process. In my opinion, this could be avoided if some data and discussion allocated in the SI in the present form of the manuscript would be moved to the main manuscript, because they are important for the understanding of the underlying mechanisms; while others could be, perhaps, moved to the SI. This is only a suggestion that I describe below.

My comments below address all these aspects in detail.

(1) Appropriateness of the control samples. Fig. 1e – G' of the Ex situ sample is smaller than G' of the mineral free sample. Later in the text you mentioned that this could be caused by topological defects. The nanoparticle volume fraction is, in fact, huge, (~ 2 vol% in Fig. 3). This is much higher

than compared to the amount of mineral in the In Situ samples, and hence, it is possible that smaller vol% could lead to much less defects and a smaller decrease in G' compared to mineral free samples, or even an increase. This makes me doubt about the appropriateness of the Ex situ sample as a control. Why did not you prepare, instead, ex situ gels with the same volume fraction of mineral as the in situ gels? I believe that a more reasonable comparison would be possible in this case, and I doubt that the current comparison is significant. I also recommend adding one sentence about why defects happen during gelation in the presence of the nanoparticles.

(2) Aggregates vs. single particles. Aggregate formation could be an artifact arising from the drying of your samples, since drying leads to significant shrinkage.

a. Can you exclude that aggregation happens due to shrinkage? Our own experience, and also reported in the literature, is that drying via ethanol-water mixtures leads to significant shrinkage and you used this method to dry the samples for TEM. Aggregates are shown in TEM images but you say that the amount is small. However, you consider the aggregates to model the SAXS data, which shows that the contribution of these aggregates is very relevant, as it is responsible for the broad peak show in the scattering intensity. Could you please compare the volume fraction obtained from TEM and SAXS? Is there an agreement? This result is missing in the SI. Did you dry the samples for SAXS and for TEM in the same way? You do not describe the preparation of the samples for SAXS, so, I am not sure if you dried these or if hydrated samples were used here (could you clarify this in the sample preparation?).

b. Also related to this, you say that the number of aggregates is small, however, your cartoons only show polycrystalline particles? Is this cartoon representative of your system?

c. I am wondering if this is affecting the mineralization in following cycles and hence it is the origin for a low increase in G' . Can you re-mineralize without intermediate drying steps? Are the results similar?

(3) Overall, I do not understand how you can exclude nucleation in other sites except at the crosslinks. I expect secondary nucleation to be relevant, especially in subsequent mineralization cycles. This is important, especially considering my concerns about the appropriateness of the control sample in (1).

(4) Estimated volume fraction of mineral. Both TEM and magnetism suggest that the vol% of mineral is smaller than the theoretical value, which is very intriguing, but the deviation is not justified.

a. Related to this, you did not explain if the mineralized hydrogels are thoroughly rinsed to remove all excess inorganic... Could this justify the smaller volume fractions?

b. On the other hand, I am wondering if the deviation is also a result of drying artifacts. Could drying cause a non-uniform distribution of the nanoparticles? This would affect the TEM results, as here, you select only a few images for the analysis. In this regard, how many images were taken for each sample? Did you analyze images "close to the surface" and further from the surface to examine if concentration gradients exist? This could justify a deviation from the theoretical vol% of mineral.

c. With regard to magnetic properties... The model used to estimate the vol% of nanoparticles is based on macroscopic magnetic properties. Does the model apply to nanoparticles as small as 1 nm? Does it include size effects? Could this be the origin for an error and the deviation from the theoretical value?

(5) Model for the mineral-induced recruitment of elastically inactive polymer network chains. I believe that the proposed model is an important mechanism that should be described in the manuscript and not in the SI. I am aware that I recommend including more information in the manuscript and there is a restriction of the space. My opinion is that it is more important to explain well the results and the underlying mechanisms in the manuscript. Instead, I would recommend placing the mineralization with histidine to the SI, as this is based on the same mechanisms. Or instead, you could move part of the methods to the SI. I believe this would improve the clarity of this manuscript.

(6) USAXS data. I do not think that Figure 2e is sufficiently informative. You could add here the data for the non-mineralized hydrogels (perhaps as inset if it is clearer). This would hinder the burden of looking for this information in the SI. Similarly, you could show the contribution of the

different particle size in this diagram.

(7) Raman spectra are taken on dry samples. How does this affect mineral-polymer interactions? Do you have controls on hydrated samples?

(8) Clarifications about the experimental methods:

- a. Did you rinse the gels after mineralization to remove the excess of chemicals? This could explain the smaller amount of mineral, but it is not explicitly said in the section,
- b. I have my doubts about the proper selection of the control Ex situ, as described earlier.
- c. For several methods you dehydrate your samples, often just exposing it to low RH, which can lead to shrinkage. Is the re-swelling reversible?
- d. Did you dry the samples for SAXS? How?

(9) Other comments:

- a. Page 5 – line 22: you mention the SEM images of the Mineral Only samples and I understand this means, absence of hydrogel. Why do you use these images to support the poor mechanical properties of the ex situ hydrogels? Why these pictures of the ex situ hydrogels support the limited network incorporation in the gels?
- b. A general comment about the photographs of the hydrogels in the manuscript. I have looked at them for some time and I do not know which information can be inferred from each of them. Can you elaborate this better?
- c. Page 5 - line 22. You say that there is no gel assembly upon mineralization within solutions of polymers without the metal-coordinating ligand and that this supports the direct role of metal coordination in controlling mineralization. But to me, the lack of gel assembly only supports that these crosslinks are needed for gelation. What am I missing here?
- d. Some of your diagrams showing G' and G'' are unclear. Can you add arrows to indicate which curves correspond to each system?

(10) Supplementary Information

- a. Figure S1: label G' (squares) and G'' (triangles)
- b. Figure S3 is mineral only samples (no gel). The last sentence of the caption says that Ex situ samples also show visual mineral precipitation as observed in Mineral Only samples. What does this mean?
- c. How did you determine $M_x=2500$ g/mol? If this is an assumption, how do you know this is a correct number?
- d. Page 6 line 11. You mention reference 17, a previous work that investigated a similar system to yours. You should also clarify the difference of your work from previous mineralization of 4cPEG gels with Fe₃O₄ nanoparticles.
- e. Diagrams with Raman spectra. Please, add a vertical line at 680 cm⁻¹ because this is an important band in your system.

Sincerely,

Rosa M. Espinosa-Marzal

Reviewer #3 (Remarks to the Author):

The manuscript by Kim et al. shows how hydrogels formed by a mixture of covalent and coordination crosslinks can be used as templates for in situ formation of minerals. The idea is interesting and novel, but there are several questions that remain unanswered (see below) that raise question both about the specific claims made in the manuscript but also to the general motivation of the work.

1. The motivation of the manuscript is to emulate the highly mineralized teeth of chitons and other highly mineralized structures. The present work, while interesting, falls quite far short of this goal. The degree of mineralization achieved is low and the stiffness in the system does not increase after the first round of mineralization indicating that the system is basically still a weakly connected particle network and not a highly mineralized composite. This should be acknowledged in the text. Also it may be beneficial to conduct experiments with higher polymer and metal loading to see if more bulk composite behavior can be attained.

2. The work of Studdart has approached biocomposite analogues from the other end of the spectrum, namely the high mineral content one. It would be worth contrasting the current approach with that one.
3. There is a vast body of literature on crystallization in hydrogels. The authors may wish to contrast their own approach with aspects of previous work – a starting point could be work of Estroff, the book *Crystals in Gels and Liesegang Rings* by Henisch, the papers by Busch/Kniep and/or the work of Imai. This comparison could serve to further underline the advantage of the presented approach in comparison to previous efforts and the general field of crystal growth in gels.
4. The materials characterization is problematic and incomplete. It is essential to provide XRD of high quality to support the claims made. The authors claim to form magnetite. I very much doubt this is the case since they form nanoparticles in air and these conditions favor the formation of maghemite instead of magnetite. The two can be told apart most easily by Mössbauer spectroscopy or by careful analysis of very high quality XRD. The same problem for the copper-gels. The claim is that copper hydroxide forms, but given the high pH, carbonate from air and the relatively low metal loading I would not be surprised if a mixture of copper hydroxide and malachite (or possibly mixed hydroxychlorides). At any rate, it is essential to characterize the minerals by XRD prior to publication – this is a minimum requirement for publication in my view and I cannot support publication without it.
5. In the abstract, the authors claim that they use “monodisperse polymers”. This is unlikely to be the case and such claims should be substantiated by detailed characterization.
6. In the introduction, I miss references to the pioneering work of Amstad/Reimhult on the use of catechols/polyphenols to stabilize iron oxide NPs
7. The speculation on page 8 that the aggregates form through oriented attachment is not substantiated by the data and should be removed or toned very significantly down.
8. The SAXS data are nice. However, the fits to them leaves something to be desired. Figure 2e clearly indicates additional smearing, which I assume is indicative of additional polydispersity not captured by simply assuming the polydispersity of the TEM data. Why was the polydispersity not explicitly fitted in the model? This should be discussed in much more detail and Table S1 should include standard deviations for all fit parameters to allow evaluating the quality of the extracted information.
9. Pertaining to the discussion of covalent crosslinking on page 12: it would be highly beneficial to estimate the amount of covalent crosslinking. This can be done easily by extracting iron with edta at low pH and conduct UV/VIS spectroscopy possibly through addition of a catechol specific dye. This point is important since the proposed mechanism is based on the assumption that a significant number of catechols remain (this is supported but not quantified by the Raman data).
10. In the discussion, the authors state that their method provides better spatial control. However, it is not clear that the method can be scaled to larger volumes since it is based on infusion of iron(II) solution into the gel. For larger gel pieces, one would expect a competition between mineralization and infusion that may inhibit loading (see also discussions in the book by Mann “*Biom mineralization*” on infusion of nanoparticles into polymeric scaffolds). This raises a number of questions, two of which are
 - a. To which degree are the authors sure that the gels formed are homogeneous?
 - b. What happens if the gel volume is increased? (important as it pertains to the potential scalability of the approach)
11. In the methods page 17, please define BOP
12. Why were tensile tests conducted at 50% of the concentration of the other gels? This information should be transferred from the methods to the main text so that the reader is aware that tensile data are on different systems

13. It is problematic that the Raman data are on dried specimens – how do you ensure that crystallization does not proceed during the drying process? Since this is the only attempt at identifying the mineral phase formed, this point is crucial.

14. The magnetization is measured on dried specimens where the nanoparticles are in closer proximity – to which degree does this impact the measured magnetization? The data look ok, but this point should be addressed.

15. There is a fair bit of scatter between the repeats in Figure S1a, please comment

16. The discussion in the supplementary material page 17 where the authors conclude that the number of elastically active chains is increased in in situ gels is at odds with the proposed model that nanoparticles exclusively form at sites already crosslinked by coordination chemistry. This is a potential problem for the central concept and should be addressed in much more detailed.

Reviewer #4 (Remarks to the Author):

This manuscript by Holten-Andersen and colleagues presents a type of metal-coordinate polymer networks reinforced by in situ crosslink mineralization. This work generally includes a versatile synthetic approach, evidence of mineral particle growth, evaluation of stiffness and magnetic performance, and generalization to different metal-ligand systems. Experimental results are systematically compared with a series of control study: ex-situ, mineral-free, ligand-free and various mineralization cycles. Overall, the manuscript is well-written. I recommend for publication with a minor revision after considering the following:

1) Figure 2b: More information should be obtained from high-resolution TEM images. What's the d-spacing of lattice fringes and which diffraction plane of Fe₃O₄ it corresponds to? Does the lattice fringe change with different mineralization cycles? More evidence can be provided to prove the formation of Fe₃O₄ particles (probably WDAX will be helpful).

2) Page 8 Line 13: Besides the volume fraction of mineral content, is there any other more accurate methods to estimate the conversion of Fe³⁺ ions to Fe₃O₄ (maybe TGA or other characterizations to obtain mass ratio)?

3) Page 11 Line 15: "The figure reveals a significant increase in the ultimate shear strength (i.e., the maximum in G') upon mineralization." Why samples In Situ × 3 and In situ × 5 didn't follow this trend (for both Figure 2h and Figure S12)?

4) Figure 4a: Does this system also contain covalent crosslink as indicated in Figure 1a?

5) Page 19 Line 5: Does it make a difference to the particle sizes, mechanical and magnetic properties by going through the mineralization process stepwise or adding excess composition for mineralization at once?

Responses to Reviewers' Comments

Title: "In situ mechanical reinforcement of polymer hydrogels via metal-coordinated crosslink mineralization"

Authors: Sungjin Kim^{†,a}, Abigail U. Regitsky^{†,a}, Jake Song^a, Jan Ilavsky^c, Gareth H. McKinley^b and Niels Holten-Andersen^{*,a}

Manuscript ID: NCOMMS-20-02410-T

We thank the reviewers for their helpful comments in revising the manuscript. Below please find our point-by-point responses.

Reviewers' comments:

Reviewer #1 (Remarks to the Author):

Holten-Andersen and coauthors demonstrate that metal-ion coordination complexes can serve as nucleation sites for in situ mineralization of hydrogels, leading to mechanical property enhancement as compared to mineral-free gels, or gels that are formed with pre-assembled mineral particles. Overall, I am impressed with the approach, experimental analysis and interpretation of results. Many groups (including that of Holten-Andersen) have studied aspects of metal-ion coordinated bonding and its effects on rheology in prior work; however, this new study shows something quite different: the ability to introduce mineralization and demonstrate its impressive impacts on mechanics, which is both innovative and exciting. I believe this will be an important contribution to the biomaterials literature. That said, there are a number of improvements that I feel would strengthen the paper, as described below:

1. The emphasis in the abstract on sustainable manufacturing seems oversold, given the current set of results which are focused more on scientific analysis than process optimization and scalability. I would reframe the study in terms of the fundamental discovery aims rather than translation to manufacturing. A brief comment on future uses in the discussion may be warranted, but the current abstract is misleading, given the results and discussion that follow. If authors prefer to keep the abstract as-written, then more details of how this manufacturing method compares to current state of the art, in terms of yield and resultant material properties is warranted, as well as a discussion of the practicality of using this for true manufacturing purposes.

RE: We thank the reviewer for the suggestion. Following the reviewer's comment, we reframed our study focusing more on the fundamental discovery by removing our statements on possible translation to manufacturing from the abstract and only briefly mentioning them in the discussion section.

2. Similarly, the introduction emphasizes magnetic mineralization using iron-containing complexes, but the authors later show that it is possible to nucleate Ni(OH)₂ and Cu(OH)₂ as well. What is the benefit of emphasizing the iron-containing clusters (particularly given the oxidation issues associated with catecholic bonding sites)? Is this simply a good method for characterization of the sample via magnetic fields, or are there other specific advantages? This should be clarified in the revision.

RE: While we were loosely inspired by the chiton that form Fe-based biominerals, our main motivation for our choice of model system is that Fe-catechol coordination is one of the most widely studied bioinspired metal-coordinate crosslinking systems with a well-established understanding of the unique mechano-chemical coupling between Fe-catechol bond equilibrium dynamics and load-bearing mechanics. Therefore, by using a combination of rheology and Raman spectroscopy we can clearly distinguish between the catechol-Fe³⁺ ion coordination state and the catechol-iron oxide nanoparticle coordination state, which is critical for identifying the in situ growth of iron oxide nanoparticles (Figure 1g, Figure S16). In addition, as the reviewer pointed out, the magnetic properties of the iron oxide minerals certainly are advantageous both for material characterization, but also for potential future material functionalization. To clarify our approach, we took out any specific mention of chiton teeth and modified the following description in the introduction as follows: “While the Fe-catechol system serves as an ideal platform to test our proof of concept of metal-coordinate crosslink mineralization^{15,23-25}, we further examined if this approach could be extended to different metal-ligand coordination systems using histidine-modified polymers which bind ions and minerals of nickel or copper.”

3. In Figure 1, the reaction path shows $2\text{Fe}^{3+} + 1\text{Fe}^{2+} + 8\text{OH}^-$, but in the caption, authors list a ratio of $\text{Fe}^{2+}:\text{Fe}^{3+}$ as 1:3. Is the Fe^{3+} added in excess? Please add more details describing how the relative ratio of ions (and other reagents) are chosen, and what (if any) effect excess agents would have.

RE: We designed the experiment based on the naïve assumption that one Fe³⁺ is forming one tris-coordinate bond with three catechol ligands (Fe³⁺:catechol = 1:3), while the remaining two Fe³⁺ are participating in the shown mineralization reaction ($2\text{Fe}^{3+} + 1\text{Fe}^{2+} + 8\text{OH}^- \rightarrow \text{Fe}_3\text{O}_4$) to produce one Fe₃O₄ per one tris-Fe-catechol crosslink site, assuming a yield of mineralization of 100%. Following the reviewer’s comment, we added this point in the methods section for clarification.

4. Is the schematic shown in Figure 1a showing an accurate ratio of covalent and metal-coordinate crosslinks, and is the distance between the various netpoints accurate?

RE: Figure 1a was purely schematic and was only intended to represent a conceptual rendering of the Mineral-Free gel with no attempt to accurately describe any molecular dimensions or composition. However, for clarity we decided to remove Figure 1a to instead focus more directly on the proposed changes to network crosslinks. In addition, we added the following statement to the figure caption: “Note that the schematic drawings in (a) – (d) are

conceptual and not meant to describe the detailed structure of the network crosslinks nor the minerals.”

5. Please add error bars to the plateau modulus data shown in Figure 1f. Is the difference observed between the mineral-free and ex situ data significant? In the discussion of these data on page 5 (lines 19-21), authors refer to Figure 1c to demonstrate precipitation in the ex situ case, but this figure shows only a schematic. Are there photographs showing visible precipitation that can be included instead or added to the SI?

RE: Following the reviewer’s suggestion, we added error bars to Figure 1f. The Ex Situ sample shows significantly diminished G_p compared to Mineral-Free, which is in agreement with the expected poor network crosslinking by the large pre-formed mineral precipitates caused by their low specific surface area compared to the smaller in situ grown nanoparticles observed in In Situ gels. Furthermore, two photographs of Ex Situ samples are now shown in Figure 1c demonstrating visible particle precipitation, alongside similar photos of Ligand-Free samples with clear particle precipitation shown in Figure 1d. For further clarification, we modified the following description in the main text: “Finally, within solutions of polymers without a metal-coordinating ligand on the backbone (Ligand-Free), as shown in Figure 1d we observed even more pronounced precipitation compared to Ex Situ, in addition to no gel assembly upon mineralization.”

6. In Figure 1g, Can the relative proportion of mineralized and complexed ions be estimated from the two peaks observed in the in situ sample?

RE: The mineralization-induced broadening of the Fe^{3+} -coordination triplet peaks around ~ 530 to $\sim 650\text{ cm}^{-1}$ supports the conversion of network crosslinks from Fe^{3+} -coordination to mineral-coordination, as supported by previously reported data with iron oxide NP crosslinked gels (Li et al., ACS Nano, 2016). In addition, the peak at $\sim 680\text{ cm}^{-1}$ indicates the formation of iron oxide mineral. Since this peak broadening appears to correlate strongly with the degree of mineralization (for example, we also observe more enhanced peak broadening in In Situ x 5 gels and less so in In Situ (1:3) gels, as shown in Figure S16), as the reviewer suggested, it would indeed be great if we could use our data set to go one step further and quantify the fraction of mineralized and complexed ions by Raman. However, since quantitative Raman analysis relies on the integration of well-resolved resonance peaks, the very nature of the peak broadening prevents a quantitative interpretation of the relative proportion of mineralized and complexed ions. Hence, we believe it is only justified to interpret the mineralization-induced Raman peak broadening qualitatively, and instead rely on our magnetic and electron microscopy measurements for quantitative estimates of sample mineral contents.

7. In Figure 2d, the in situ x4 and x5 data sets do not show clear saturation at the maximal applied H-fields. Can authors increase $|H|$ to better determine the maximal value? In Figure S6, authors indicate that they can extract a maximal value from a linear extrapolation of $1/H$ versus M but do not show this data or include the full equation to which the data are fit. These should be included. Also, why at low cycle numbers (x2, x3) do the volume fractions determined via M_s value and measured maximal value not agree, given the obvious saturation in the raw data at high $|H|$?

RE: Please note that the previous Figure S6 has now been reassigned to Figure S7. As the reviewer suggested, increasing the $|H|$ range was the first idea we had. However, even at an H range of ± 70000 Oe and a run of 6+ hours per sample, we found a negligible change in our sample output and therefore no change for our overall interpretations. Following the reviewer's suggestion, we now added the extrapolation fits for all 5 samples as a new Figure S8. As seen from this Figure, even though the In Situ x 2, x 3 may appear to be saturated in magnetization in Figure 2d, they did in fact not saturate in their linear extrapolation from $1/H$ vs M graphs at high $|H|$ (i.e., $1/|H| \rightarrow 0$) in contrast to the In Situ x 1. Thus, we used the intercepts of the linear extrapolation to estimate the M_s of our samples except for the In Situ x 1.

8. Did authors measure the magnetic properties of the ex situ mineralized gels? If so, what properties did they show? If not, why not?

RE: Since the purpose of the Ex Situ gels was to test the predicted poor integration of mineral particles grown under uncontrolled conditions, we focused on characterizing their bulk mechanical properties and their polymer-particle interface via Raman (as shown in Figure 1). While Ex Situ gels should respond to an external magnetic field, analysis of this data would not provide us with the same direct insights on the efficiency of particle-polymer network coupling similar to what we can extract from rheology and Raman data. We agree with the reviewer, that magnetic characterization of Ex Situ gels could be interesting and useful in future work, especially work more specifically focused on testing the possibly enhanced efficiency of magnetic actuation of gels with magnetic particles grown in situ.

9. For the tensile testing data shown in Figure S11b – how many samples were tested for each condition? Are the error bars showing standard deviation from replicate measurements? What was the hydration state of the gels during tensile testing? If different than the hydration state of the gels during LAOS, how would you expect these changes in wetting to influence the results?

RE: Please note that this Figure has been reassigned to Figure S13b. Each condition was tested with 3 independent samples and the error bars are standard deviations of these triplicate experiments. This information has been added to the figure S13 caption. Each test was done on freshly prepared hydrated samples to minimize the influence of dehydration.

10. For the LAOS data shown in Figure 2h, is there a meaning to the shear strain at which the peak value in strength occurs? Also, the caption indicates that the strain energy density increases with mineralization – how is this determined from the plots shown in Panel 2h?

RE: The curve of $|G' \cdot \gamma|$ vs γ can be treated somewhat as an analogue to a stress-strain curve. The shear strain (γ) at the maximum stress can, therefore, be interpreted as a measure of the yield strain, since the maximum shear strength is defined by $\max |G' \cdot \gamma|$ and G' stays constant until initiation of sample failure, which results in a drop in G' . The strain energy density (J/m^3) is the area underneath the $|G' \cdot \gamma|$ vs γ curve, which in literature occasionally has been treated analogously to toughness. However, we refrained from using the term “toughness”, since it is usually reserved for the property determined by more classical tensile-fracture tests. Since the scope of the current paper was instead to investigate the change in bulk rheological properties

of gels upon in situ crosslink mineralization, we chose to reserve attempted measurements of mineral-induced enhanced fracture toughness to future studies.

11. In Figure 3, authors show the change in plateau modulus +/- mineral addition. Can authors please include the actual values or G_p before and after addition in the SI? Please also include error bars on the plot in Figure 3, propagating the uncertainty properly for the subtraction operation. Why is no magnetization data shown for the conventional NP gels?

RE: Following the reviewer's suggestion, we added a new table (Table S2) with the actual values. Error bars are now also included in Figure 3, reflecting the uncertainty for the subtraction as suggested. The magnetization data of conventional NP gels was not reported in the work we used as the reference (Li et al., ACS Nano, 2016), but should correspond to the fraction of incorporated pre-synthesized Fe_3O_4 NPs (i.e., ~1.97 vol.%, which is significantly higher than the In Situ gels in our report). As stated above for Ex Situ gels, magnetic characterization of conventional NP gels could be interesting and useful in future work, especially work more specifically focused on testing the possibly enhanced efficiency of magnetic actuation of gels with magnetic particles grown in situ.

12. Authors claim that the increased topological defects in the conventional NP case undermine the mechanical reinforcement effects these should have. Presumably the likelihood of defect formation increases with NP density. Do NP-seeded networks at volume fractions similar to those obtained with the in situ gels show better mechanical enhancement? How does the response of these gels compare to that of the in situ gels under similar NP volume fractions?

RE: Since analogous conventional NP gels require a particle loading of at least 1 vol% for gels to form (i.e. for the polymer network to percolate) (Li et al., ACS Nano, 2016), and the highest mineral loading fraction of In Situ gels we achieve in this study is ~0.37 vol%, we are unable to directly compare the full viscoelastic properties of the two composite gel systems. However, in a separate (still to be published) study we found that incorporation of NPs into an Fe-catechol coordination network indeed decreases G_p , as shown in the data below (please note that NPX indicates the NP wt% in the gel, so that 2.5 wt% of Fe_3O_4 NPs corresponds to 0.48 vol%, for example).

13. I don't understand the data in Figure S15. The schematic suggests that there are no catechol groups at the polymer ends – is this correct? If so, what is attracting the Fe^{3+} to the polymer end tri-junction? The caption suggests that there are catechols present but that the Fe^{3+} is added at low concentration to form nominally monovalent stoichiometries. Can you please revise either the text or schematic to clarify? Also, is the last sentence of the caption "...regardless of

the presence of catecholic covalent crosslinks...” correct? Does this mean that catechols are not required for the mineralization or stiffening effect? Something seems inconsistent here.

RE: Please note that this Figure has been now reassigned to Figure S17. As the reviewer pointed out, catechols are indeed required for the mineralization and stiffening effect, and we now added catechol groups to the polymer ends in the schematic figure. We thank the reviewer for pointing out this omission. In the last sentence of the caption, we meant to infer that the low concentration of Fe^{3+} induce an insignificant level of catechol-oxidation and thereby a low level of catecholic covalent crosslinks in the network. We have now clarified this statement in the figure caption as follows: “The low concentration of Fe^{3+} induce an insignificant level of catechol-oxidation and thereby a low level of catecholic covalent crosslinks in the network. These observations thus support that the in situ mineralization stiffen the network, regardless of the fraction of catecholic covalent crosslinks in the system.”

14. The plateau modulus data in Figure 4 suggests that the ion identity has a large effect on the mechanical behavior. Can the authors please add either a brief explanation or appropriate references to other work to explain this effect? In general, is the fold increase between the mineral-free and in situ data meaningful (i.e., can this be reconciled with the single-ion binding affinity and/or and local concentration of mineral NP in the gel)?

RE: The seminal work by Fullenkamp et al. on the metal-histidine coordinate crosslinked hydrogels was already referenced in our original submission, but this work has now been emphasized by stating that: “Importantly, unlike the oxidation-prone catechol-based gel networks above, these histidine-metal coordinate networks do not contain covalent crosslinks, hence they behave close to purely transient viscoelastic fluids before any network mineralization, as has been reported in previous work⁴²⁻⁴⁴.” Furthermore, in the main text we now included a discussion of a proposed mineralization-induced reinforcing mechanism based on classical rubber elasticity theory of crosslink efficiency and functionality which was previously located in the Supplementary Information. Since we are not changing the species of ion nor ligands upon triggering mineralization, we propose that crosslink efficiency and functionality, rather than single-ion binding affinity, constitute the major cause of the observed mineral-induced reinforcement mechanism.

Reviewer #2 (Remarks to the Author):

The manuscript by Kim et al. describes how significant mechanical reinforcement of hydrogels can be achieved by in situ metal coordinated crosslink mineralization. This work describes a series of carefully performed experiments that characterize the mineralized hydrogels (and controls) made of PEG end-functionalized with metal-coordinating ligands of either catechol or histidine. The authors propose that the nucleation of the mineral nanoparticles happens in the crosslinks via metal coordination, which yields to significant reinforcement. In my opinion, this paper deserves publication in Nature Communication and congratulate the authors for the beautiful results. However, there are some parts of the manuscript that require clarification before publication, which, I wish, will further strengthen this manuscript and its implications. I recognize that the manuscript contains a large amount of experimental data, and that many of them need to be in the SI. As a result, reading the manuscript is not a fluent process. In my opinion, this could be avoided if some data and discussion allocated in the SI in the present form of the manuscript would be moved to the main manuscript, because they are important for the understanding of the underlying mechanisms; while others could be, perhaps, moved to the SI. This is only a suggestion that I describe below.

My comments below address all these aspects in detail.

(1) Appropriateness of the control samples. Fig. 1e – G' of the Ex situ sample is smaller than G' of the mineral free sample. Later in the text you mentioned that this could be caused by topological defects. The nanoparticle volume fraction is, in fact, huge, (~2 vol% in Fig. 3). This is much higher than compared to the amount of mineral in the In Situ samples, and hence, it is possible that smaller vol% could lead to much less defects and a smaller decrease in G' compared to mineral free samples, or even an increase. This makes me doubt about the appropriateness of the Ex situ sample as a control. Why did not you prepare, instead, ex situ gels with the same volume fraction of mineral as the in situ gels? I believe that a more reasonable comparison would be possible in this case, and I doubt that the current comparison is significant. I also recommend adding one sentence about why defects happen during gelation in the presence of the nanoparticles.

RE: We thank the reviewer for this comment. To clarify, the Ex Situ gels in Figure 1 are indeed prepared with exactly the same composition as In Situ gels. Hence, the Ex Situ and In Situ gels are expected to contain the same volume % of minerals by design. Their difference instead lies in the order of processing; in Ex Situ gels mineral nucleation and growth is initiated prior to mixing with the metal-coordinating polymers, whereas in In Situ gels mineral nucleation and growth takes place within the already established metal-coordinate polymer network.

On the other hand, the Conventional NP gels introduced in Figure 3 are prepared by mixing identical metal-coordinating polymers with carefully presynthesized Fe_3O_4 NPs (with particle sizes comparable to those observed in our In Situ gels) in a process that required at least 1 vol% of NPs and 24 hours of curing for gelation (Li et al., ACS Nano, 2016). Nonetheless, this is a well established protocol of NP composite hydrogel assembly, which is the reason we

compared the particle-induced mechanical reinforcement of such “Conventional NP gels” with our In Situ gels in Figure 3.

Finally, both Ex Situ and Conventional NP gels made by mixing-in particles show a decrease in their stiffness compared with their mineral-free (In Situ) counterpart. Indeed, we agree with the reviewer that a higher NP vol% will decrease G_p . In fact, in a separate (still to be published) study we already confirmed that conventional incorporation of presynthesized NPs into an Fe-catechol coordination network decreases G_p , as shown in the data below (Please note that NPX indicates the NP wt% in the gel, so that 2.5 wt% of Fe_3O_4 NPs corresponds to 0.48 vol%, for example).

Hence, in both Ex Situ and Conventional NP gels, an increase in NP vol. % decreases G_p , in contrast to our in situ mineralization approach, which does not result in such compromise in the gel modulus (Figure 3). Also following the reviewer’s final recommendation, we added the following statement in the main text: “In contrast, despite a tenfold higher content of Fe_3O_4 nanoparticles (~ 2 vol%), a similar analysis of a conventional nanoparticle gel shows a decrease of ~5000 Pa in gel stiffness, a common effect explained by the introduction of elastically inactive polymer chains such as loops, and its associated decrease in network elasticity, upon conventional mixing in of pre-synthesized nanoparticles in gels^{15,19–21}.”

(2) Aggregates vs. single particles. Aggregate formation could be an artifact arising from the drying of your samples, since drying leads to significant shrinkage.

a. Can you exclude that aggregation happens due to shrinkage? Our own experience, and also reported in the literature, is that drying via ethanol-water mixtures leads to significant shrinkage and you used this method to dry the samples for TEM. Aggregates are shown in TEM images but you say that the amount is small. However, you consider the aggregates to model the SAXS data, which shows that the contribution of these aggregates is very relevant, as it is responsible for the broad peak show in the scattering intensity. Could you please compare the volume fraction obtained from TEM and SAXS? Is there an agreement? This result is missing in the SI. Did you dry the samples for SAXS and for TEM in the same way? You do not describe the preparation of the samples for SAXS, so, I am not sure if you dried these or if hydrated samples were used here (could you clarify this in the sample preparation?).

RE: We thank the reviewer for this inquiry. For USAXS, all of the measurements were conducted on hydrated samples, which is clarified in the methods section. For TEM, while some sample volume change caused by the solvent exchange process is expected, the sample was always kept in a swollen state with respective solvents and any observed volume change was not significant. We added this note to the methods section. Indeed, to make sure the shrinkage was not a significant influence, we crosschecked the volume fractions calculated from the TEM images and magnetic measurements. They match well with each other.

Measuring the volume fraction of our 5x gel via USAXS is challenging due to the uncertainty in the contrast of the grown particles relative to the matrix and the solvent. These can be affected by the presence of unaccounted Fe^{3+} ions or amorphous $\text{Fe}_3\text{O}_4/\text{Fe}_2\text{O}_3$ in the gel solvent, all of which would affect the contrast and thus impact the calculated volume fraction of the gel. It is for these reasons we had originally decided to use the volume fraction/contrast terms in Equation S1 as scaling parameters.

However, we do recognize the intent of the question by the reviewer, and have therefore attempted to calculate the volume fraction of the gel (Fig. R1 shown below). For this calculation, we use the same fitting parameters for small spheres, large spheres, and cylinders as before (Table. S1), but fix the scattering length density to that of ideal Fe_3O_4 ($41.67 \times 10^{10} \text{cm}^{-2}$) vs water ($9.14 \times 10^{10} \text{cm}^{-2}$) resulting in scattering contrast between the materials of $[(41.67 - 9.14) \times 10^{10} \text{cm}^{-2}]^2 = 1040 \times 10^{20} \text{cm}^{-4}$. We assume that the larger aggregates are constituted by the large spheres, and therefore the volume fraction of the aggregates is inherently accounted for by the volume fraction of the large spheres. Furthermore, we subtract the contribution from the 4-arm polymer / mineral-free in the fitting process, which we do by estimating the contribution of the mineral-free gel (Fig. S10) with a sphere model, and including this contribution into the current fitting process (this is a necessary step to accurately estimate the small sphere populations since their sizes are comparable to the 4-arm polymers themselves).

The resulting volume fractions for this fitting procedure (denoted Fit 1 in Fig. R1) are $\varphi_{\text{small sphere}} = 0.035 \%$, $\varphi_{\text{large sphere}} = 0.06 \%$, $\varphi_{\text{cylinders}} = 0.03 \%$, summing to $\varphi_{\text{mineral}} = 0.125\%$. This is considerably smaller than the TEM estimate of $\varphi_{\text{mineral}} = 0.29\%$ but we believe that the agreement is actually quite good, considering the significant difference in length-scales between the two measurements, and not having accounted for possible Fe^{3+} ions or amorphous $\text{Fe}_3\text{O}_4/\text{Fe}_2\text{O}_3$ in the gel which will underestimate the USAXS volume fraction (if it were possible to account for these elements, the actual contrast of the minerals would be lower, thus the volume fraction required to fit the USAXS data would be higher).

We believe that aggregates are made up of the large spheres. Quantifying the volume of aggregates using SAXS technique is very difficult since the contrast of aggregates is highly uncertain. Therefore, we make a simplified assumption that the aggregates are composed only of extra-large spheres and fit it as such (denoted Fit 2 in Fig. R1). In this case we obtain $\varphi_{\text{extra-large spheres}} = 0.045 \%$ - this does provide some insight into the relative volume fraction of the aggregates, which would indeed be fairly significant. This would raise the total volume fraction to $\varphi_{\text{mineral}} = 0.17 \%$. We stress, however, that this is a simplification and a convenient assumption enabling us to model (with a reasonable number of parameters) this complicated system.

In either case, we believe that the aggregates are significant and real, even in the hydrated gel specimen, as shown by the USAXS results. Our observation agrees with prior literature which suggests that such aggregation processes are in fact typical of magnetite nucleation and growth in solution (Baumgartner et al., Nat. Mater. 2013).

Fig. R1. Fitting of the USAXS data using absolute contrast values of Fe_3O_4 . Fit 1 is the modified USAXS fit including small sphere, large sphere, and cylinder, using the absolute contrast of Fe_3O_4 .

b. Also related to this, you say that the number of aggregates is small, however, your cartoons only show polycrystalline particles? Is this cartoon representative of your system?

RE: The schematic cartoons in Figure 1 are conceptual, and we did not intend to depict the actual structure of the polymer network nor the minerals within. We have now clarified this point in the figure caption. Yet, we do believe conceptually illustrating our system this way is appropriate since to our knowledge, it is much easier to form polycrystalline (composed of many crystallites of varying size and orientation) particles than to form single crystalline particles, especially in the simple mineral formation processes we use. We note that “Aggregates” is a term we used loosely to distinguish particles larger than what we designated as “Large” particles to accurately extract particle statistics from the TEM images (Figure 2c, Figure S4, S5) to be used for USAXS modeling, and we do not mean to imply that individual “Large” or “Small” particles are all single crystals.

c. I am wondering if this is affecting the mineralization in following cycles and hence it is the origin for a low increase in G' . Can you re-mineralize without intermediate drying steps? Are the results similar?

RE: As the reviewer proposed, we ideally would have performed the additional mineralization cycles without intermediate drying. However, without intermediate drying, it would not have been possible to precisely control the effective gel concentration of the reactive re-mineralizing species, which we deemed to be essential for accurate and reliable comparisons between cycles. However, we do not expect that the required dehydration steps artificially suppresses the

increase in gel modulus, since in a separate study we are currently exploring how to mineralize a gel directly in a bulk aqueous mineralization bath, and in this study we have also witnessed a saturation in the gel modulus (please see figure below displaying G_p of a weakly crosslinked mineral-free nitrocatechol network as a function of continuous mineralization time in a bulk bath).

Related to the possible concern of dehydration causing a change in the mineralization and the network property, we added a control test comparing a fresh-made gel and rehydrated gel in Figure S18 and its caption as follows: “Comparison of viscoelastic properties measured by rheological frequency sweep of In Situ $\times 2$ (magenta), In Situ $\times 3$ (purple) and In Situ $\times 2$ rehyd (skyblue) that went through free dehydration over 24 hours in air then rehydrated with the equivalent original volume. The difference caused by dehydration is negligible (i.e., In Situ $\times 2$ vs In Situ $\times 2$ rehyd) compared to that caused by an additional mineralization cycle (i.e., In Situ $\times 2$ vs In Situ $\times 3$), suggesting that the dehydration is not the major cause of the mineralization to induce the solidification of the material.”

(3) Overall, I do not understand how you can exclude nucleation in other sites except at the crosslinks. I expect secondary nucleation to be relevant, especially in subsequent mineralization cycles. This is important, especially considering my concerns about the appropriateness of the control sample in (1).

RE: We completely agree with the reviewer’s insight. Indeed, we do not exclude the possibility of particle formation outside the crosslinks. This point was previously discussed in the Supporting Materials but is now included in the main text stating: “... while we cannot rule out possible particle nucleation outside the coordinate crosslink sites, our Raman (Supplementary Figure 16) and mechanical data (Figure 3, Supplementary Figure 17, 18) suggest that later mineralization cycles result in further growth of the particles already formed, and it is tempting to speculate if mineral-bound catechol ligands become entrapped in growing particles⁴¹ during these additional cycles of mineralization. Such ligand-mineral entrapment could potentially both explain the lack of increased moduli and the slower stress relaxation of more mineralized

gels, as well as the diminishing Raman resonance for Fe³⁺-catechol coordination (~530 to ~650 cm⁻¹) and increase in the peak for iron oxides (~680 cm⁻¹) observed after five cycles of mineralization (Supplementary Figure 16).”

(4) Estimated volume fraction of mineral. Both TEM and magnetism suggest that the vol% of mineral is smaller than the theoretical value, which is very intriguing, but the deviation is not justified.

a. Related to this, you did not explain if the mineralized hydrogels are thoroughly rinsed to remove all excess inorganic... Could this justify the smaller volume fractions?

RE: Following the reviewer’s comment, we have now clarified in the methods section that any excess ions are indeed removed by carefully rinsing and dabbing the gels after each cycle. We also included in the supporting information (Figure S7) the reviewer’s suggestion that this could be a factor justifying the smaller volume fractions as follows: “**While there could be some product loss from the sample washing step,** we note that the Φ_{magnetic} obtained from the magnetization analysis treats all minerals as magnetite, which thereby discounts any possible volume fraction of less magnetic side products or intermediates.”

b. On the other hand, I am wondering if the deviation is also a result of drying artifacts. Could drying cause a non-uniform distribution of the nanoparticles? This would affect the TEM results, as here, you select only a few images for the analysis. In this regard, how many images were taken for each sample? Did you analyze images “close to the surface” and further from the surface to examine if concentration gradients exist? This could justify a deviation from the theoretical vol% of mineral.

RE: As explained above in our response to (2)a., the TEM measurements were performed on samples fixed in their swollen states. Yet, we agree that the vol% estimated by TEM measurements is imperfect since we cannot completely rule out shrinkage during sample prep (possibly leading to gradient distributions of mineral particles as suggested by the reviewer), and we did not perform a type of Z-stack analysis with our TEM sections to check for such gradients. However, 10 images were taken for the In Situ x1 and 40 images were taken for the In Situ x5 sample which we used for calculating the mineral volume fraction (please see Table 1 in methods section and Figure S4, S5). In addition, we also compared these TEM-based estimates with both magnetically-based measurement and theoretical calculations to validate the TEM analysis. Although we did not investigate specifically the possible existence of a hydrogel mineral gradient in our current TEM studies, we are actively seeking funding to perform more in-depth studies of diffusion-limited mineral gradients in metal-coordinate hydrogels.

c. With regard to magnetic properties... The model used to estimate the vol% of nanoparticles is based on macroscopic magnetic properties. Does the model apply to nanoparticles as small as 1 nm? Does it include size effects? Could this be the origin for an error and the deviation from the theoretical value?

RE: We deeply appreciate the reviewer’s input on this matter. Following the reviewer’s comment, in the caption to Figure S7 we have added that very small-sized particles may have

a much lower number of domains and magneto-crystalline anisotropy, which can respond differently to an external magnetic field. This is in addition to our original comment in Figure S7 that the existence of impurities other than magnetites could also cause the deviation. However, since our gels did exhibit superparamagnetic behaviors, we believe spontaneous magnetization caused by such magnetic anisotropy was insignificant in our study.

(5) Model for the mineral-induced recruitment of elastically inactive polymer network chains. I believe that the proposed model is an important mechanism that should be described in the manuscript and not in the SI. I am aware that I recommend including more information in the manuscript and there is a restriction of the space. My opinion is that it is more important to explain well the results and the underlying mechanisms in the manuscript. Instead, I would recommend placing the mineralization with histidine to the SI, as this is based on the same mechanisms. Or instead, you could move part of the methods to the SI. I believe this would improve the clarity of this manuscript.

RE: Following the reviewer's comment, we have now included the discussion on the proposed mechanism for mineral-induced recruitment of elastically inactive polymer network chains in the main text instead of the SI. We are still below the 5000 words limit for the main text (not including abstract, Methods, References and figure legends), so we do not need to place the histidine-functionalized polymer gel mineralization study in the SI.

(6) USAXS data. I do not think that Figure 2e is sufficiently informative. You could add here the data for the non-mineralized hydrogels (perhaps as inset if it is clearer). This would hinder the burden of looking for this information in the SI. Similarly, you could show the contribution of the different particle size in this diagram.

RE: We thank the reviewer for this suggestion and we added the non-mineralized data as an inset to the current Figure 2e. However, we believe this combined plot is more appropriate in the SI (Figure S10) for specialist readers to read both plots concurrently.

(7) Raman spectra are taken on dry samples. How does this affect mineral-polymer interactions? Do you have controls on hydrated samples?

RE: We also performed Raman spectroscopy on hydrated samples, however except for an overall reduction in the signal intensity due to the dilution of the sample, no shifts in the Raman spectra were observed between the hydrated (left figure below) and dehydrated state (right figure below) of an In Situ x5 gel sample. Since the Raman laser dries out the target position

of the gel sample very quickly, we deem that reporting the Raman data of a sample dried under controlled conditions is more appropriate.

(8) Clarifications about the experimental methods:

a. Did you rinse the gels after mineralization to remove the excess of chemicals? This could explain the smaller amount of mineral, but it is not explicitly said in the section,

RE: Please see our answer to (4)a. above.

b. I have my doubts about the proper selection of the control Ex situ, as described earlier.

RE: Please see our answer to (1) above.

c. For several methods you dehydrate your samples, often just exposing it to low RH, which can lead to shrinkage. Is the re-swelling reversible?

RE: Yes, as explained above in our answer to (2c), upon gel rehydration we found that there was no significant difference in the viscoelastic behavior between samples before dehydration and after rehydration.

d. Did you dry the samples for SAXS? How?

RE: No, the samples remained hydrated for SAXS measurements. We have now clarified this point in the methods section.

(9) Other comments:

a. Page 5 – line 22: you mention the SEM images of the Mineral Only samples and I understand this means, absence of hydrogel. Why do you use these images to support the poor mechanical properties of the ex situ hydrogels? Why these pictures of the ex situ hydrogels support the limited network incorporation in the gels?

RE: We thank the reviewer for this comment. We agree that our original direct reference to the SEM images of mineral precipitates from Mineral-Only samples as an example of the heterogeneous state of mineral particles in Ex Situ gels was not representative. The point we wanted to make was simply that the Ex Situ gel is formed by mixing the solution of the pre-formed mineral precipitates (as in Mineral-Only, characterized in Supplementary Figure 3) with the polymer solution to induce gelation. We have now eliminated the direct reference to the SEM images of mineral precipitates and instead simply increased the size of the gel images in Figure 1. Hence the proposed poor network integration leading to poor gel formation, can now more easily be deduced from the visible precipitation in Ex Situ gels. In addition to increasing the gel image size, we also clarified the accompanying main text as follows: “In contrast, with uncontrolled mineral nucleation and growth initially outside the

metal-coordinating network, Ex Situ gels displayed comparatively poor mechanical properties, likely due to limited network incorporation of the pre-formed large and heterogeneous particles visibly precipitating in the sample as shown in Figure 1c. Note that the Ex Situ gel is formed by mixing a solution of pre-formed mineral precipitates (as in Mineral-Only, further characterized in Supplementary Figure 3) with the polymer solution to induce gelation.” In addition, we edited in the Figure S3 caption for further clarification as follows: “We note that the Ex Situ gel samples are processed via first forming a dispersion of mineral precipitates (as shown in inset photo) by mixing all ingredients without 4cPEG, followed by later mixing with 4cPEG polymer solution.”

b. A general comment about the photographs of the hydrogels in the manuscript. I have looked at them for some time and I do not know which information can be inferred from each of them. Can you elaborate this better?

RE: Following the reviewer’s comment, as mentioned above, we have now increased the size of the gel images in Figure 1, and hence the proposed poor network integration leading to poor gel formation and no gel formation in Ex Situ and Ligand-Free, respectively, can now more easily be deduced from the visible precipitation in the images of the samples. In addition, new descriptions are included in the Figure 1 caption: “Note the visible mineral particle precipitation in both the Ex Situ and Ligand-Free samples ... Note that the Ex Situ samples only display weak gel-like behavior and the Ligand-Free samples are liquids in agreement with the qualitative appearance of the samples in (c) and (d).” In addition, we added the following description to the Figure 4 caption: “Inset images are photographs of each sample. Note that precipitation of particles is visible in the Ex Situ gels.” We also modified the relevant text in the main text for clarification; please see our answer on the next question Q9c.

c. Page 5 - line 22. You say that there is no gel assembly upon mineralization within solutions of polymers without the metal-coordinating ligand and that this supports the direct role of metal coordination in controlling mineralization. But to me, the lack of gel assembly only supports that these crosslinks are needed for gelation. What am I missing here?

RE: Related to our answer to Q9b, for clarification, we modified our description as follows: “Finally, within solutions of polymers without a metal-coordinating ligand on the backbone (Ligand-Free), as shown in Figure 1d we observed even more pronounced precipitation compared to Ex Situ, in addition to no gel assembly upon mineralization. These observations combined support a direct role of metal-coordination in controlling crosslink mineralization as well as negligible contributions from mineral-polymer interactions outside the network crosslink sites to both network formation and elasticity.”

We believe it is important that when particles are grown in solutions of polymers without metal-coordinate crosslinking sites there is no signs of any network formation (i.e. no gel is formed in the Ligand-Free sample), which suggests only weak and shortlived interactions between particles and the polymer backbone. In addition, from representative macroscopic images now shown in the SI, Fig. S3, the extent of visible mineral precipitation in Mineral-Only samples looks similar to the Ligand-Free samples (representative image shown in Fig.1). Combined, these findings suggest that any particles grown outside the crosslink sites in In Situ gels do not

contribute significantly to the network elasticity and that the metal-coordination exerts some level of control over mineralization resulting in the observed well-distributed nanoparticles.

d. Some of your diagrams showing G' and G'' are unclear. Can you add arrows to indicate which curves correspond to each system?

RE: Following the reviewer's comment, we clarified Figures 1 and 4 by matching the colors of the labels with the plots and adding arrows where needed.

(10) Supplementary Information

a. Figure S1: label G' (squares) and G'' (triangles)

RE: We thank the reviewer for pointing this out and we have added the labels.

b. Figure S3 is mineral only samples (no gel). The last sentence of the caption says that Ex situ samples also show visual mineral precipitation as observed in Mineral Only samples. What does this mean?

RE: Related to our answers to Q.9a-c above, we meant that the precipitates are visually seen before mixing with the polymer to form the Ex Situ (Figure 1c). For clarification, we modified the sentence in Figure S3 as follows: "We note that the Ex Situ samples are formed via mixing in these pre-formed precipitates with 4cPEG polymer solution."

c. How did you determine $M_x=2500$ g/mol? If this is an assumption, how do you know this is a correct number?

RE: The 4-arm-PEG (MW=10kDa) backbone polymer we used is sold as such, thus we believe it is reasonable to assume that the length of each arm is on average are 2500 g/mol. For clarification, we modified our description in the supporting information as follows: (molecular weight between crosslink junctions, i.e., average 1 arm length of our 10kDa 4-arm-PEG backbone polymer).

[Redacted]

Example of a commercial 4-arm-PEG backbone from JenKem Technology:

<https://www.jenkemusa.com/product/4arm-peg-succinimidyl-carboxymethyl-ester>

d. Page 6 line 11. You mention reference 17, a previous work that investigated a similar system to yours. You should also clarify the difference of your work from previous mineralization of 4cPEG gels with Fe_3O_4 nanoparticles.

RE: Please refer to our answer on the Question 1, where we tried to clarify this concern.

e. Diagrams with Raman spectra. Please, add a vertical line at 680 cm^{-1} because this is an important band in your system.

RE: This is a great suggestion, and we added this line, alongside other vertical lines, to highlight additional important Raman bands in the spectrum.

Reviewer #3 (Remarks to the Author):

The manuscript by Kim et al. shows how hydrogels formed by a mixture of covalent and coordination crosslinks can be used as templates for in situ formation of minerals. The idea is interesting and novel, but there are several questions that remain unanswered (see below) that raise question both about the specific claims made in the manuscript but also to the general motivation of the work.

1. The motivation of the manuscript is to emulate the highly mineralized teeth of chitons and other highly mineralized structures. The present work, while interesting, falls quite far short of this goal. The degree of mineralization achieved is low and the stiffness in the system does not increase after the first round of mineralization indicating that the system is basically still a weakly connected particle network and not a highly mineralized composite. This should be acknowledged in the text. Also it may be beneficial to conduct experiments with higher polymer and metal loading to see if more bulk composite behavior can be attained.

RE: We thank the reviewer for this comment, and we agree that our original introductory paragraphs overemphasized our inspiration from biological materials such as chiton teeth. In fact, our motivation for this work was not to produce solid-like composites mimicking highly mineralized biological materials such as chiton teeth. Rather, we simply attempted to explore the efficiency of network strengthening via targeted crosslink mineralization generally inspired by the spatio-temporal in situ scaffold mineralization well documented in such biological material stiffening processes. Following the reviewer's comment, we modified the introduction section to clarify our scope, and specifically we deleted our opening statements regarding chiton teeth. Furthermore, we added the following statement at the end of the paper in the discussion to emphasize our limited scope: "We note that the scope of this paper was not to mimic hard-condensed biological composite materials such as nacre^{52,53} or chiton teeth^{3,54}. Rather we focused on a fundamental exploration of a new bio-inspired approach to reinforcement of organic-inorganic soft-condensed matter that could prove advantageous and efficient compared to conventional routes."

2. The work of Studdart has approached biocomposite analogues from the other end of the spectrum, namely the high mineral content one. It would be worth contrasting the current approach with that one.

RE: We thank the reviewer for this comment and accordingly, we included the following statement citing the recommended authors' works (refs 52, 53) in the discussion: "Mineralization in macromolecular hydrogel networks is a broad field of study which have focused on various important topics such as mineral morphogenesis control⁴⁵⁻⁴⁹, mineral incorporation of macromolecules^{41,50,51} as well as the influence of mineralization on mechanical properties, for example of solid nacre-like nanocomposites^{52,53}."

3. There is a vast body of literature on crystallization in hydrogels. The authors may wish to contrast their own approach with aspects of previous work – a starting point could be work of Estroff, the book Crystals in Gels and Liesegang Rings by Henisch, the papers by Busch/Kniep

and/or the work of Imai. This comparison could serve to further underline the advantage of the presented approach in comparison to previous efforts and the general field of crystal growth in gels.

RE: We thank the reviewer for this comment and we appreciate the opportunity to further emphasize the vast body of literature on crystal growth in gels, and more clearly articulate how we believe our work contributes to this wide field. Accordingly, we included the following statement with suggested works (refs 45-51) in the discussion related to our answer to Q2: “Mineralization in macromolecular hydrogel networks is a broad field of study which have focused on various important topics such as mineral morphogenesis control⁴⁵⁻⁴⁹, mineral incorporation of macromolecules^{41,50,51} as well as the influence of mineralization on mechanical properties, for example of solid nacre-like nanocomposites^{52,53}”

4. The materials characterization is problematic and incomplete. It is essential to provide XRD of high quality to support the claims made. The authors claim to form magnetite. I very much doubt this is the case since they form nanoparticles in air and these conditions favor the formation of maghemite instead of magnetite. The two can be told apart most easily by Mössbauer spectroscopy or by careful analysis of very high quality XRD. The same problem for the copper-gels. The claim is that copper hydroxide forms, but given the high pH, carbonate from air and the relatively low metal loading I would not be surprised if a mixture of copper hydroxide and malachite (or possibly mixed hydroxychlorides). At any rate, it is essential to characterize the minerals by XRD prior to publication – this is a minimum requirement for publication in my view and I cannot support publication without it.

RE: We deeply appreciate this insight offered by the reviewer. Indeed, XRD was the first characterization technique we tried to identify the mineral in our gel networks. However, as shown below, the diffraction pattern from the small fraction of minerals in the gels was too weak to be clearly distinguished from the signal stemming from the PEG polymer itself.

XRD pattern of In Situ gels. The most prominent peak for (311) plane of magnetites or maghemites at 35.42° or 35.63°, respectively, coincides with a peak originating from the PEG polymer (please compare with the pure PEG XRD pattern).

XRD pattern of pure PEG (red, blue) from Fu, X., et al., 2016. *Chemical Engineering Journal*, 291, pp.138-148. Note: PCM stands for Phase Change Materials with PEG base in their study.

However, as an alternative to XRD, we instead performed HRTEM analysis to quantify the d-spacing of the minerals as grown directly in our In Situ hydrogel (newly added Figure S6) and found that the d-spacings correspond better with those reported of magnetites, although magnetites and maghemites share the same crystal family (which again makes the distinction between the two phases difficult using XRD). We also note that we have now expanded our deductive power of Raman spectroscopy to not only allow us to characterize the state of the catechol-mineral particle interface coordination, but additionally now also in our attempt to determine the identity of the minerals grown in the network. A reference by Hanesch (*Hanesch, Geophys. J. Int.*, 2009) now cited in the manuscript explicitly states that "Raman spectroscopy is an easy method to distinguish magnetite and maghemite.", and thanks to the insight from the reviewer, we indeed found a trace of the distinguishing Raman peak of maghemite (i.e., $\sim 730 \text{ cm}^{-1}$) in addition to the stronger band around 680 cm^{-1} indicative of magnetite in our existing Raman spectrum of the In Situ gel displayed in Figure 1 of the main manuscript. Both of these bands have now been highlighted in the figure and the accompanying figure caption.

Therefore, we believe it is reasonable to regard that our systems include magnetites as well as maghemite. Furthermore, we did not intend to deny the existence of additional "impurities", since inducing the pure magnetite phase was not the scope of our work. We also appreciate and agree with the reviewer's point on the possible inclusion of other forms of minerals other than metal hydroxides in our histidine systems, yet, due to the same problem of the strong XRD peak of PEG, we found it difficult to explicitly define the mineral we have induced in our hydrogel systems using XRD. We added the reviewer's points on the mineral identity in the main text and supplementary information for clarification.

5. In the abstract, the authors claim that they use "monodisperse polymers". This is unlikely to be the case and such claims should be substantiated by detailed characterization.

RE: We agree with the reviewer's comment. Accordingly, we deleted the claim of polymer monodispersity from the manuscript.

6. In the introduction, I miss references to the pioneering work of Amstad/Reimhult on the use of catechols/polyphenols to stabilize iron oxide NPs

RE: Following the reviewer's comment, we added the recommended works (Amstad et al., Nano Lett. 2009; J. Phys. Chem. C, 2011; Nanoscale, 2011) as **references 16-18**.

7. The speculation on page 8 that the aggregates form through oriented attachment is not substantiated by the data and should be removed or toned very significantly down.

RE: Following the reviewer's comment, we deleted the corresponding statement.

8. The SAXS data are nice. However, the fits to them leaves something to be desired. Figure 2e clearly indicates additional smearing, which I assume is indicative of additional polydispersity not captured by simply assuming the polydispersity of the TEM data. Why was the polydispersity not explicitly fitted in the model? This should be discussed in much more detail and Table S1 should include standard deviations for all fit parameters to allow evaluating the quality of the extracted information.

RE: We thank the reviewer for this observation. We agree that the SAXS fits in Figure 2e are not perfect and that additional smearing by an increased size distribution would improve the match of fit to the data. However, our concern stems from the fact that the multi-population model based on TEM images has already too many parameters, and that the information contained in the SAXS data is limited, resulting in a number of not unique solutions to more complicated models. Therefore, with a system as complex as ours, it would be easy to over-fit and over-interpret the data.

We, therefore, have chosen to limit the number of fitting parameters and accept some misfits, but make sure the fitting results are trustworthy, and based on microscopic information obtained from other measurements such as TEM. In turn, this shows that our TEM images, which are derived from nanometer-scale sample sizes, do a reasonably good job in representing the microstructure of the macroscopic sample, evidenced by the good agreement with the USAXS measurement which is taken at a millimeter-scale on a fully-hydrated specimen.

9. Pertaining to the discussion of covalent crosslinking on page 12: it would be highly beneficial to estimate the amount of covalent crosslinking. This can be done easily by extracting iron with edta at low pH and conduct UV/VIS spectroscopy possibly through addition of a catechol specific dye. This point is important since the proposed mechanism is based on the assumption that a significant number of catechols remain (this is supported but not quantified by the Raman data).

RE: We appreciate this suggestion by the reviewer. In particular because we in previous studies indeed estimated the degree of covalent crosslinking in identical Fe-4cPEG networks using a similar method of dissolving out the non-permanent coordinate network fraction by iron

extraction with EDTA, followed by mass-based estimates of the remaining covalent network fraction (S. Kim et al., Chem. Mater. 2018). We agree that this information is pertinent to the discussion on the possible contribution of covalent crosslinks to the In Situ mineralized gels, and therefore we added this point in the supporting information (Figure S17) as follows: “These observations thus support that the in situ mineralization stiffen the network, regardless of the fraction of catecholic covalent crosslinks in the system (e.g., ~33 wt % for Mineral-Free gels based on estimates of the gel mass fraction after dissolving out the transient metal-coordinate crosslinked fraction of the gel network as reported in S. Kim et al., Chem. Mater. 2018²⁰).” Please note that we decided to move this detailed discussion on the covalent fraction from the main text to Figure S17, since we instead relocated the detailed discussion on in situ mineralization mechanisms to the main text.

10. In the discussion, the authors state that their method provides better spatial control. However, it is not clear that the method can be scaled to larger volumes since it is based on infusion of iron(II) solution into the gel. For larger gel pieces, one would expect a competition between mineralization and infusion that may inhibit loading (see also discussions in the book by Mann “Biom mineralization” on infusion of nanoparticles into polymeric scaffolds). This raises a number of questions, two of which are

a. To which degree are the authors sure that the gels formed are homogeneous?

RE: This is an important point raised by the reviewer. To be clear, when we state that our approach of network crosslink mineralization via metal-coordination provides better spatial control, what we specifically propose is that we have improved the level of control over where in the polymer network the mineral particles nucleate and grow (i.e. the metal-coordinate complex crosslinks). We completely agree with the reviewer that the resulting distribution of the grown mineral particles is still under the constraints of diffusion following the infusion of iron(II) solution into the gel. However, while there may indeed be a small spatial distribution of particles due to the diffusion-limited infusion of Fe^{2+} , to the best of our effort we did not detect any signs of large and steep mineral gradients neither in our ultra-structural analysis (please see representative TEM images in Fig. 2b and 2c), nor in our mechanical analysis (due to sample geometry, rheology is particularly sensitive to mechanical gradients, which works in our favor in this instance). This lack of a detectable mineral gradient we believe is due to a combination of our small sample sizes (as the reviewer points out) and our sample prep being carefully designed to try to avoid gel heterogeneity (please see Methods section for details of sample prep).

b. What happens if the gel volume is increased? (important as it pertains to the potential scalability of the approach)

RE: In a separate ongoing project, we are indeed exploring how to achieve homogeneous in situ mineralization in larger gel sample sizes. By inducing gel mineralization in a bulk bath, we are able to allow much longer time scales for infusion of ions and thereby ensure homogeneous distribution of minerals over longer length scales. Although we are still optimizing this mineralization bath method, we are hopeful it will prove itself useful as one possible method with which to scale up our in situ mineralization approach. Crude preliminary

test results show that the gel stiffening effect remains significant, as can be seen in the following videos of large-size samples before and after in situ mineralization.

11. In the methods page 17, please define BOP

RE: Following the reviewer's comment, we defined BOP (benzotriazol-1-yloxytris(dimethylamino)phosphonium hexafluorophosphate) in the methods section.

12. Why were tensile tests conducted at 50% of the concentration of the other gels? This information should be transferred from the methods to the main text so that the reader is aware that tensile data are on different systems

RE: We had to prepare the tensile test specimens at a 0.5x diluted sample concentration to slow down the mineralization reaction kinetics to reliably produce dogbone-shaped samples without obvious gel heterogeneities. We clarified this in the methods section as well as in the supporting information (Figure S13) and made a specific reference to this clarification in the main text.

13. It is problematic that the Raman data are on dried specimens – how do you ensure that crystallization does not proceed during the drying process? Since this is the only attempt at identifying the mineral phase formed, this point is crucial.

RE: Out of the same concern we have also tried Raman spectroscopy on hydrated samples. However, except for an overall reduction in the signal intensity due to the dilution of the sample, we did not observe a significant difference between the hydrated (left figure below) and dehydrated sample state (right figure below). Additionally, because the Raman laser dries out the target position of the sample very quickly, we decided to report our Raman data on dehydrated samples. Finally, as mentioned in our response to reviewer comment #4 above, we have now also explored the identity of the mineral phase with XRD and HRTEM.

To further test if gel dehydration had a major influence on mineralization, we also tried a different approach. After rehydrating a dehydrated gel back to its original volume we compared its rheological properties with a fresh hydrated gel, and we found no significant difference in the viscoelastic behavior between the original hydrated and the rehydrated samples. Given the sensitivity of rheology to the state of network mineralization, this finding serves as further support that our Raman data on dry gels presented in the main manuscript is indeed representative. We added this new rheological evidence as Figure S18 with its caption: **“Comparison of viscoelastic properties measured by rheological frequency sweep of In Situ x**

2 (magenta), In Situ × 3 (purple) and In Situ × 2 rehyd (skyblue) that went through free dehydration over 24 hours in air then rehydrated with the equivalent original volume. The difference caused by dehydration is negligible (i.e., In Situ × 2 vs In Situ × 2 rehyd) compared to that caused by an additional mineralization cycle (i.e., In Situ × 2 vs In Situ × 3), suggesting that the dehydration is not the major cause of the mineralization to induce the solidification of the material.”

14. The magnetization is measured on dried specimens where the nanoparticles are in closer proximity – to which degree does this impact the measured magnetization? The data look ok, but this point should be addressed.

RE: We thank the reviewer for this suggestion. Following the reviewer’s comment, we added the following description to Figure S7: “Furthermore, since we used dehydrated instead of hydrated gels to prevent sample volume change during the multi-hour measurement, it is likely that the dehydration shortened interparticle distances in the gel, which could possibly form extrinsic magnetic anisotropy thereby inducing spontaneous magnetism. However, based on the superparamagnetic behavior lacking a coercivity in our dehydrated gels, such events were negligible to affect the magnetic properties.”

15. There is a fair bit of scatter between the repeats in Figure S1a, please comment

RE: We thank the reviewer for this comment. We rechecked the value and one of the four samples indeed displayed a significantly lower G_p than the three other samples. We believe that the aberrant behavior of this sample resulted from a smaller initial addition of Fe^{3+} . Thus, we have removed this aberrant plot from Figure S1a.

16. The discussion in the supplementary material page 17 where the authors conclude that the number of elastically active chains is increased in in situ gels is at odds with (conflict with) the proposed model that nanoparticles exclusively form at sites already crosslinked by coordination chemistry. This is a potential problem for the central concept and should be addressed in much more detailed.

RE: We thank the reviewer for this input. While we cannot rule out that dangling chains can nucleate and grow particles, we believe that the primary mechanism of recruitment of elastically active chains is based upon nucleation and growth of particles in the metal-coordinate crosslinks. While we admit that the mechanism is largely based on our speculation, our perspective stems from previous works suggesting that more than three chains can be bound to the nanoparticle as potential crosslinkers – i.e., they have higher functionalities (Li et al., ACS Nano, 2016; Amstad et al., Nano Lett. 2009). As requested by the reviewer, we have now elaborated more on this argument in the main text by placing the following discussion on the proposed recruitment mechanism: “Specifically, the observation that only the first cycle of mineralization causes a significant increase in gel modulus could plausibly be explained by the transformation of initial low functionality bis- or tris-catechol- Fe^{3+} coordinate complexes into high functionality catechol-mineral crosslink structures through the recruitment of polymer network chains initially elastically inactive in the Mineral-Free gel during the process of nucleation and growth of particles directly in the

metal-coordinate crosslinks (see Supplementary Figure 15 for more in depth discussion on this mechanism)^{39,40}. Furthermore, while we cannot rule out possible particle nucleation outside the coordinate crosslink sites, our Raman (Supplementary Figure 16) and mechanical data (Figure 3, Supplementary Figure 17, 18) suggest that later mineralization cycles result in further growth of the particles already formed, and it is tempting to speculate if mineral-bound catechol ligands become entrapped in growing particles⁴¹ during these additional cycles of mineralization. Such ligand-mineral entrapment could potentially both explain the lack of increased moduli and the slower stress relaxation of more mineralized gels, as well as the diminishing Raman resonance for Fe³⁺-catechol coordination (~530 to ~650 cm⁻¹) and increase in the peak for iron oxides (~680 cm⁻¹) observed after five cycles of mineralization (Supplementary Figure 16).”

Hence, we believe that it is reasonable to propose that mineralization possibly recruits elastically inactive chains to crosslink-mineralization sites since it is known that the nanoparticle crosslinkers can have significantly higher functionalities. We do not deny possible mineral formation outside the metal-coordinate crosslink sites, however we not believe this would contradict our model that the nanoparticles form primarily at sites of metal-coordinate crosslinks, which again is supported by separate Raman measurements (Figure S16) and mechanical tests (Figure 3).

Reviewer #4 (Remarks to the Author):

This manuscript by Holten-Andersen and colleagues presents a type of metal-coordinate polymer networks reinforced by in situ crosslink mineralization. This work generally includes a versatile synthetic approach, evidence of mineral particle growth, evaluation of stiffness and magnetic performance, and generalization to different metal-ligand systems. Experimental results are systematically compared with a series of control study: ex-situ, mineral-free, ligand-free and various mineralization cycles. Overall, the manuscript is well-written. I recommend for publication with a minor revision after considering the following:

1) Figure 2b: More information should be obtained from high-resolution TEM images. What's the d-spacing of lattice fringes and which diffraction plane of Fe_3O_4 it corresponds to? Does the lattice fringe change with different mineralization cycles? More evidence can be provided to prove the formation of Fe_3O_4 particles (probably WDAX will be helpful).

RE: We thank the reviewer for this comment. The d-spacing of the particle from an In Situ x5 gel shown in the inset of Figure 2c in the main manuscript is 0.299 nm, which correlates with the (220) plane of magnetite. Following the reviewer's comment, we added Figure S6 which include multiple d-spacing observations in further agreement with the proposed formation of magnetite. We tried similar analysis in In Situ x1 gels but we were only able to reliably measure the well-defined lattice fringes in "Aggregates" in In Situ x 5 gels. We also tried WAXD on In Situ x 1 gels. However, the diffraction pattern from such a small fraction of minerals in the PEG gel could not be distinctively resolved from the intrinsic PEG signal (please see the XRD pattern below).

XRD pattern of In Situ gels. The most prominent peak for (311) plane of magnetites or maghemites at 35.42° or 35.63° , respectively, coincides with a peak originating from the PEG polymer (please compare with the pure PEG XRD pattern).

XRD pattern of pure PEG (red, blue) from Fu, X., et al., 2016. *Chemical Engineering Journal*, 291, pp.138-148. Note: PCM stands for Phase Change Materials with PEG base in their study.

However, as an alternative to XRD, we instead performed HRTEM analysis to quantify the d-spacing of the minerals as grown directly in our In Situ hydrogel (newly added Figure S6) and found that the d-spacings correspond better with those reported of magnetites, although magnetites and maghemites share the same crystal family (which again makes the distinction between the two phases difficult using XRD). We also note that we have now expanded our deductive power of Raman spectroscopy to not only allow us to characterize the state of the catechol-mineral particle interface coordination, but additionally now also in our attempt to determine the identity of the minerals grown in the network. A reference by Hanesch (Hanesch, *Geophys. J. Int.*, 2009) now cited in the manuscript explicitly states that "Raman spectroscopy is an easy method to distinguish magnetite and maghemite.", and thanks to the insight from the reviewer, we indeed found a trace of the distinguishing Raman peak of maghemite (i.e., $\sim 730 \text{ cm}^{-1}$) in addition to the stronger band around 680 cm^{-1} indicative of magnetite in our existing Raman spectrum of the In Situ gel displayed in Figure 1 of the main manuscript. Both of these bands have now been highlighted in the figure and the accompanying figure caption.

2) Page 8 Line 13: Besides the volume fraction of mineral content, is there any other more accurate methods to estimate the conversion of Fe^{3+} ions to Fe_3O_4 (maybe TGA or other characterizations to obtain mass ratio)?

RE: This is a great question. We indeed tried TGA but found it difficult to assign which mass drop corresponds to which mineral phase since heating itself can affect the mineralization of un-mineralized ions and change the phase of the Fe minerals. While we have not found a more accurate quantitative method to estimate the conversion of Fe^{3+} ions to Fe_3O_4 than using volume fractions of the minerals in the hydrogel for this paper, this is one of the major reasons we relied on Raman spectroscopy to validate the identity of the mineral in the network as well as the state of the catechol-metal coordination in our mineralized gels. We are however

currently searching for other quantitative methodologies with which to estimate the conversion of ions to minerals in future studies.

3) Page 11 Line 15: “The figure reveals a significant increase in the ultimate shear strength (i.e., the maximum in $G'\gamma$) upon mineralization.” Why samples In Situ $\times 3$ and In situ $\times 5$ didn't follow this trend (for both Figure 2h and Figure S12)?

RE: Please note that the previous Figure S12 has been reassigned to Figure S14. This figure shows moduli (stiffness), not strength, and similar to what we observed from G_p in linear rheological tests shown in Fig. 2f, the tensile moduli after the first mineralization cycle likewise do not differ significantly. Our point was simply that the strength, as well as G_p , increased significantly from the unmineralized sample to any of the mineralized samples. We do not yet have a clear understanding to speculate on the variation in mechanical behavior between the mineralized samples.

4) Figure 4a: Does this system also contain covalent crosslink as indicated in Figure 1a?

RE: There are no covalent crosslinks in any of the histidine gel networks depicted in Figure 4, i.e. they are completely transiently crosslinked. Furthermore, due to other reviewers' remarks, we decided to delete the previous conceptual depictions of gel networks (previous Figure 4a and Figure 1a) for simplicity.

5) Page 19 Line 5: Does it make a difference to the particle sizes, mechanical and magnetic properties by going through the mineralization process stepwise or adding excess composition for mineralization at once?

RE: Adding all the “ingredients” at once to match the final In Situ $\times 5$ conditions in our current multi-step-cycle In Situ gels would no doubt make a difference in the outcome. Specifically, using such concentrated ingredients all at once would change the initial supersaturation of ions, thus the nucleation and growth rate would change exponentially, which would make mineral formation rate, final particle size, magnetic and mechanical properties different from what we have observed from our step-wise approach to the In Situ $\times 5$ mineralization level. We thank the reviewer for this comment since it is indeed an interesting idea to explore, but one we believe would go beyond the current scope of the paper.

REVIEWERS' COMMENTS

Reviewer #1 (Remarks to the Author):

The authors have addressed my concerns, and I have no further comments or suggestions for changes.

Reviewer #2 (Remarks to the Author):

The authors have clarified all my questions and addressed my concerns. They have made appropriate corrections to the manuscript. Therefore, I recommend publication of this work.
Rosa Espinosa-Marzal

Reviewer #3 (Remarks to the Author):

The authors have improved the manuscript, but a few comments/questions remain. Once these are addressed, I am happy with this manuscript and strongly support its publication.

The key claim in the paper in the introduction "We found that metal-ion coordination complexes can indeed serve as direct mineral nucleation sites, whereby significant mechanical reinforcement is achieved upon nanoscale particle growth directly at the metal-coordinate network crosslink sites" is partially supported by the data, but in my view additional experiments are required to fully establish it. Indeed the authors now acknowledge that secondary nucleation events can occur. They would have to conduct control experiments that show this statement to be true. This is hard but first steps could include infusing Ni into the Fe(III)-catechol gels where one must assume that the stronger binding of Fe(III) would exclude direct catechol mediated nucleation of the nickel hydroxide. Another control could be to form the network with Al(III) and the infuse Fe(II), again gels with significantly different properties should result if the statement is correct. Additionally, if the above statement was correct, one would assume that the crystals be distributed at specific distances within the sample governed by the network crosslink density – this does not appear to be the case, at least not strongly so, as indicated by the lack of strong and sharp interparticle correlation peaks in the USAXS data – or it should be clearly visible in e.g. cryo TEM. The possibility of secondary nucleation should be discussed in the main manuscript and not relegated to supplementary information. Either stronger experimental evidence for the strong claim in the introduction should be provided or the statement softened a bit – I suggest the latter.

The maghemite vs magnetite discussion remains muddled. The use of lattice fringes in high resolution TEM images to distinguish the two is wrought with danger – these data are in my view not strong enough for conclusions. I would suggest that the authors either go the full way (Mössbauer) or just acknowledge that there most likely is a mixture of phases. It detracts nothing from the work at all (incidentally, storage in air will most likely increase the proportion of maghemite over time).

The USAXS data: did you fit all data (in situ x1 – 5)? Only data on in situ x5 models are provided. Please show data (and fits) for wall samples to allow following the mineralization process.

The In situ mineralized Cu-histidine system appear synergetic in the picture in Figure 4d – was it? The XRD data from the rebuttal should be included in the supplementary information to demonstrate that you tried XRD and that the particle concentration is too low to allow drawing conclusions.

Reviewer #4 (Remarks to the Author):

Although some of concerns are not fully addressed (probably due to the nature of some mechanisms and structures of these materials), overall I am satisfied with the responses from the authors. Thus, I am happy to support for acceptance.

Responses to Reviewers' Comments

Title: "In situ mechanical reinforcement of polymer hydrogels via metal-coordinated crosslink mineralization"

Authors: Sungjin Kim^{†,a}, Abigail U. Regitsky^{†,a}, Jake Song^a, Jan Ilavsky^c, Gareth H. McKinley^b and Niels Holten-Andersen^{*,a}

Manuscript ID: NCOMMS-20-02410A

We thank the reviewers for their helpful comments in revising the manuscript. Below please find our point-by-point responses.

Reviewers' comments:

Reviewer #1 (Remarks to the Author):

The authors have addressed my concerns, and I have no further comments or suggestions for changes.

Reviewer #2 (Remarks to the Author):

The authors have clarified all my questions and addressed my concerns. They have made appropriate corrections to the manuscript. Therefore, I recommend publication of this work.
Rosa Espinosa-Marzal

Reviewer #3 (Remarks to the Author):

The authors have improved the manuscript, but a few comments/questions remain. Once these are addressed, I am happy with this manuscript and strongly support its publication.

The key claim in the paper in the introduction "We found that metal-ion coordination complexes can indeed serve as direct mineral nucleation sites, whereby significant mechanical reinforcement is achieved upon nanoscale particle growth directly at the metal-coordinate network crosslink sites" is partially supported by the data, but in my view additional experiments are required to fully establish it. Indeed the authors now acknowledge that secondary nucleation events can occur. They would have to conduct control experiments that show this statement to be true. This is hard but first steps could include infusing Ni into the Fe(III)-catechol gels where one must assume that the stronger binding of Fe(III) would exclude direct catechol mediated nucleation of the nickel hydroxide. Another control could be to form the network with Al(III) and the infuse Fe(II), again gels with significantly different properties should result if the statement is correct. Additionally, if the above statement was correct, one would assume that the crystals be distributed at specific distances within the sample governed by the network crosslink density – this does not appear to be the case, at least not strongly so,

as indicated by the lack of strong and sharp interparticle correlation peaks in the USAXS data – or it should be clearly visible in e.g. cryo TEM. The possibility of secondary nucleation should be discussed in the main manuscript and not relegated to supplementary information. Either stronger experimental evidence for the strong claim in the introduction should be provided or the statement softened a bit – I suggest the latter.

RE: We appreciate the reviewer's insightful input and suggestion. Following the reviewer's advice, we softened the introductory statement as follows: Here, we introduce **our findings supporting that** metal-ion coordination complexes **can** indeed serve as direct mineral nucleation sites, whereby significant mechanical reinforcement is achieved upon nanoscale particle growth directly at the metal-coordinate network crosslink sites.

In addition, the following statement in the main manuscript addresses the possibility of secondary nucleation: Furthermore, while we cannot rule out possible particle nucleation outside the coordinate crosslink sites, our Raman (Supplementary Figure 16) and mechanical data (Figure 3, Supplementary Figure 17, 18) suggest that later mineralization cycles result in further growth of the particles already formed, and it is tempting to speculate if mineral-bound catechol ligands become entrapped in growing particles⁴¹ during these additional cycles of mineralization.

The maghemite vs magnetite discussion remains muddled. The use of lattice fringes in high resolution TEM images to distinguish the two is wrought with danger – these data are in my view not strong enough for conclusions. I would suggest that the authors either go the full way (Mössbauer) or just acknowledge that there most likely is a mixture of phases. It detracts nothing from the work at all (incidentally, storage in air will most likely increase the proportion of maghemite over time).

RE: We appreciate this suggestion. Following the reviewer's advice, we acknowledged that multiple phases of iron oxides could be present in our system in the main text as follows: **We note however that a mixture of iron oxide phases is likely to result from our mineralization process and continued oxidation in air.**

The USAXS data: did you fit all data (in situ x1 – 5)? Only data on in situ x5 models are provided. Please show data (and fits) for wall samples to allow following the mineralization process.

RE: Based on our particle classification from the TEM image analysis, we ended up collecting and fitting USAXS data for x 5 only, since we believe this sample would provide the most relevant information on mineralized network structure. We agree with the reviewer that in future studies it would indeed be interesting to follow the process more closely with additional scattering analysis at various timepoints of network mineralization.

The In situ mineralized Cu-histidine system appear synergetic in the picture in Figure 4d – was it?

RE: When assembling the metal-coordinate gels, we typically do observe an initial gel contraction before the sample ingredients have been fully mixed and the sample has reached equilibrium. Since this was a freshly made gel, it is plausible that the picture was taken before the sample had time to equilibrate. However, we note that all measurements of gels were performed after sample equilibration, and that we have never observed signs of syneresis over longer time scales.

The XRD data from the rebuttal should be included in the supplementary information to demonstrate that you tried XRD and that the particle concentration is too low to allow drawing conclusions.

RE: Following the reviewer's suggestion, we now included the XRD data as Supplementary Figure 6e.

Reviewer #4 (Remarks to the Author):

Although some of concerns are not fully addressed (probably due to the nature of some mechanisms and structures of these materials), overall I am satisfied with the responses from the authors. Thus, I am happy to support for acceptance.